# SCALING LAWS OF ROPE-BASED EXTRAPOLATION

**Xiaoran Liu[1,2]**[*] **Hang Yan[1]**[*] **Chenxin An[2], Xipeng Qiu[2],**[†] **Dahua Lin[1]**

[1]Shanghai AI Lab

[2]School of Computer Science, Fudan University

liuxr22@m.fudan.edu.cn, {yanhang, lindahua}@pjlab.org.cn,
{cxan20, xpqiu}@fudan.edu.cn

## ABSTRACT

The extrapolation capability of Large Language Models (LLMs) based on Rotary Position Embedding (Su et al., 2021) is currently a topic of considerable interest. The mainstream approach to addressing extrapolation with LLMs involves modifying RoPE by replacing 10000, the rotary base of $\theta_n = 10000^{-2n/d}$ in the original RoPE, with a larger value and providing longer fine-tuning text. In this work, we first observe that fine-tuning a RoPE-based LLM with either a smaller or larger base in pre-training context length could significantly enhance its extrapolation performance. After that, we propose ***Scaling Laws of RoPE-based Extrapolation***, a unified framework from the periodic perspective, to describe the relationship between the extrapolation performance and base value as well as tuning context length. In this process, we also explain the origin of the RoPE-based extrapolation issue by ***critical dimension for extrapolation***. Besides these observations and analyses, we achieve extrapolation up to 1 million context length within only 16K training length on LLaMA2 7B and 13B (Touvron et al., 2023b).

## 1 INTRODUCTION

Large Language Models (LLMs) have become the dominant architecture in a variety of natural language processing tasks(OpenAI, 2023; Touvron et al., 2023a;b), while Transformers (Vaswani et al., 2017) based on Rotary Position Embedding (RoPE) (Su et al., 2021) have become the dominant backbone in wide range of LLM design (Chowdhery et al., 2022; Nijkamp et al., 2023; Touvron et al., 2023a;b). While RoPE can theoretically represent sequences through trigonometric functions, as detailed in Appendix A, its performance drops when the input sequence or context length surpasses the training length(Press et al., 2022; Chen et al., 2023), seen in Figure 1. This *extrapolation problem* (Press et al., 2022) limits tasks like long text modeling and summarization(An et al., 2023).

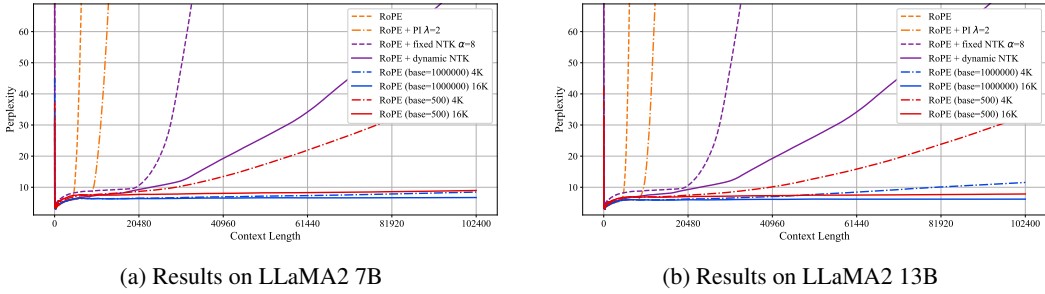

(a) Results on LLaMA2 7B · (b) Results on LLaMA2 13B

Figure 1: Perplexity on Books3 (Presser, 2020) with different extrapolation methods, including Dynamic NTK (bloc97, 2023b). RoPE fine-tuned with a smaller or larger base on the original training length of 4K or a much longer context of 16K, could outperform other extrapolation strategies and extrapolate to 100K context length.

---

[*] Equal contribution.

[†] Corresponding author: xpqiu@fudan.edu.cn

Concerning the extrapolation issue with RoPE, different works have provided various interpretations and corresponding solving attempts. These works could divided into two schools of thought. One limits the scope of self-attention (Ratner et al., 2022; Han et al., 2023) given the fact that RoPE-based self-attention fails to keep stable beyond training context and exhibits attention score explosion as well as monotonous entropy increase (Chen et al., 2023; Han et al., 2023). The other aims to capture longer contexts with smaller rotation angles and longer fine-tuning context (Chen et al., 2023; Peng et al., 2023). Current popular methods, such as Dynamic NTK (bloc97, 2023a) and Code LLaMA (Rozière et al., 2023), mainly come from the second approach. Both approaches adapt RoPE to longer contexts with a larger rotary base. Specifically, Dynamic NTK (bloc97, 2023a) adjusts the base with a coefficient increasing with the length of inference, allowing RoPE-based LLMs to adapt simultaneously to longer context, while Code LLaMA (Rozière et al., 2023) directly sets the base at 1000000 and gets further trained on 16K context length, yielding a context beyond 100K.

While recent studies have shown promising results, they have primarily focused on specific base values and tuning context lengths. This leaves a notable gap in understanding how base value, tuning length, and extrapolation performance relate. For instance, while larger bases improve extrapolation in models like LLaMA2 (Touvron et al., 2023a), surprisingly, we also find that fine-tuning with smaller bases with the original training length is also conducive to the extrapolation capability of LLaMA2, which is also demonstrated in Figure 1. Furthermore, when trained in a longer context, RoPE with a smaller base can match or even surpass those with a larger one. At the same time, fine-tuning with a base of 1000000 on the original training length also achieves extrapolation up to 100K. These findings pose several questions. ***Q1: Is 10000 the worst base value for extrapolation in the fine-tuning phase? Q2: Is there a mathematical relationship between rotary base, training context length, and extrapolation limit? Q3: If so, can we achieve unbound extrapolation accordingly?***

In this paper, we conduct further experiments on increasing and decreasing the rotary base in Section 2 and subsequently discover that adjusting the rotary base in both directions can contribute to the extrapolation of RoPE-based LLMs. Building upon these observations, we provide a comprehensive explanation for the seemingly counter-intuitive phenomenon from a periodic perspective. Meanwhile, we establish a unified theoretical framework for RoPE-based extrapolation known as the ***Scaling Laws of RoPE-based Extrapolation***[1]. We pinpoint specific changes during base reduction that lead to a significant boost in extrapolation in Section 3.1 and identify the upper bound of extrapolation for larger bases in Section 3.3. This clarifies how Code LLaMA (Rozière et al., 2023) manages a 100K extrapolation with only 16K training context. Furthermore, we validate our theories in Section 3.4 and Appendix C, shedding light on both the core principles of Dynamic NTK (bloc97, 2023a) and pinpointing instability sources self-attention computations in RoPE-based extrapolation (Han et al., 2023). Finally, we present the contributions and guiding significance of this work for other methods that achieve extrapolation during the inference phase. In summary, our contributions are as follows, and codes are available at `https://github.com/OpenLMLab/scaling-rope`.

- We first highlight a surprisingly strange phenomenon 10000 is the worst base value for RoPE-based extrapolation in the fine-tuning phase. Remarkably, fine-tuning with either a larger or smaller base within the training context length greatly enhances extrapolation, which provides a new vision to the extrapolation research of RoPE (Su et al., 2021).
- Then we introduce a unified theoretical framework for RoPE-based extrapolation from a periodic perspective, known as the *Scaling Laws of RoPE-based Extrapolation*, which not only clarifies the aforementioned observations and addresses unanswered questions in existing research (Rozière et al., 2023), but also discover the *Critical Dimension for RoPE-based Extrapolation*, revealing the underlying reasons for the extrapolation issue of RoPE.
- Finally, for extrapolation within a defined context, we present the suggested fine-tuning base value determined by the context limit and extend the context of LLaMA2 7B and 13B (Touvron et al., 2023b) to surpass 100K tokens by tuning RoPE with base 1000000 and 4K tuning length. For unpredictable extrapolation, we propose a RoPE with a smaller base, such as 500, and achieve an almost 1M token context with a mere 16K tuning length.

---

[1] It is important to note that the scaling laws proposed in this work are *irrelevant* with the well-known scaling laws (Kaplan et al., 2020). In this paper, *scale* refers to the adjustment of RoPE's rotary angles (bloc97, 2023a), rather than the change of the model size. Unlike the scaling laws (Kaplan et al., 2020) that are empirically derived, our scaling laws define mathematical relations between context window size and rotary base, supported by experiments.

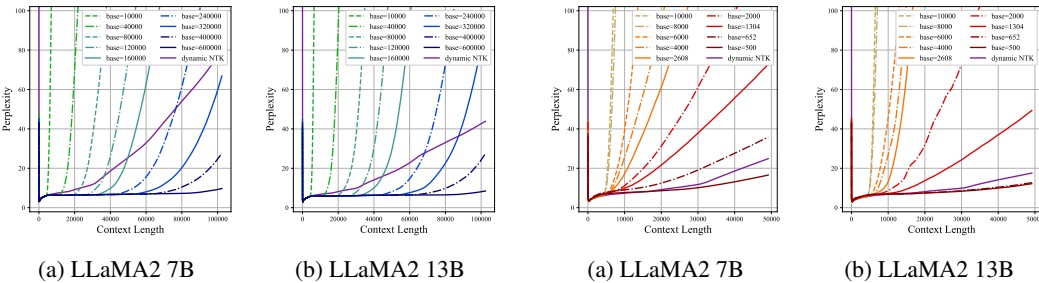

| (a) LLaMA2 7B | (b) LLaMA2 13B | (a) LLaMA2 7B | (b) LLaMA2 13B |

Figure 2: Perplexity of larger bases on Books3 (Presser, 2020) shows better extrapolation.

Figure 3: Perplexity of smaller bases on Books3 (Presser, 2020) shows better extrapolation.

## 2 OBSERVATION

### 2.1 LARGER BASES PROMISE BETTER EXTRAPOLATION

We first conduct the extrapolation experiments with larger bases, based on the experimental setup in Appendix B.1. It is evident that tuning with larger bases could significantly improve the extrapolation performance of RoPE as shown in Figure 2. Besides, there are several noteworthy points.

First, Larger bases allow LLaMA2 (Touvron et al., 2023b) to extrapolate beyond its training context length, aligning with findings from Rozière et al. (2023). Secondly, the extrapolation with larger bases has a clear limit where language modeling perplexity stays consistent. Beyond this limit, the extrapolation performance declines significantly. Furthermore, as the base value rises, LLaMA2 can extrapolate to a longer context. Finally, compared to Dynamic NTK (bloc97, 2023a), RoPE tuned with larger bases degrades much quicker beyond its extrapolation upper bound. Therefore, for fine-tuning with larger bases, the performance beyond the upper bound could be consistently overtaken by Dynamic NTK. Nevertheless, within the upper bound, this approach still outperforms Dynamic NTK by a considerable margin, leading to a context beyond 100K with only a 4K tuning length, when the base is set over 600000.

### 2.2 SMALLER BASES ALSO PROMISE BETTER EXTRAPOLATION

We then conduct the extrapolation experiments with smaller bases, using the same setup as for larger bases. Interestingly, even though this goes against common research findings (bloc97, 2023a; Rozière et al., 2023), fine-tuning RoPE with smaller bases on the original context length still boosts extrapolation, as shown in Figure 3. It also extends the context window beyond the training length. Yet, there are distinct differences when comparing RoPE with smaller bases to larger ones.

Firstly, RoPE with smaller bases does not have a distinct upper bound of extrapolation. While perplexity worsens as context length grows, this decline is gentler with smaller bases. Secondly, the enhancement in extrapolation of RoPE with smaller bases is not uniform. Between a base of 10000 and 8000, the extrapolation performance exhibits a tiny improvement. Then between a base of 8000 and 2608, improvement is moderate. After that, from 2608 to 1304 and further to 652, the improvement becomes more pronounced. Finally, when the rotary base is 500, the extrapolation curve becomes sufficiently smooth thus resulting in strong extrapolation over 48K context length and superior performance over Dynamic NTK (bloc97, 2023a).

Combining these two results, we observe a surprisingly strange phenomenon in RoPE-based extrapolation as depicted in Figure 4. Specifically, base 10000 yields the worst extrapolation performance when fine-tuned, thus answering Q1 in the Introduction. As the base either decreases or increases, performance notably improves. Interestingly, the improvements differ between the two directions. For larger bases, although the performance steadily improves, there exists a clear extrapolation upper bound. In contrast, for smaller bases, while the improvement is not uniform, the resulting extrapolation curve does not have an obvious breaking point. In the heatmap in Figure 4, there is a clear and continuous boundary for larger bases as well as a distinct transition phase for smaller bases.

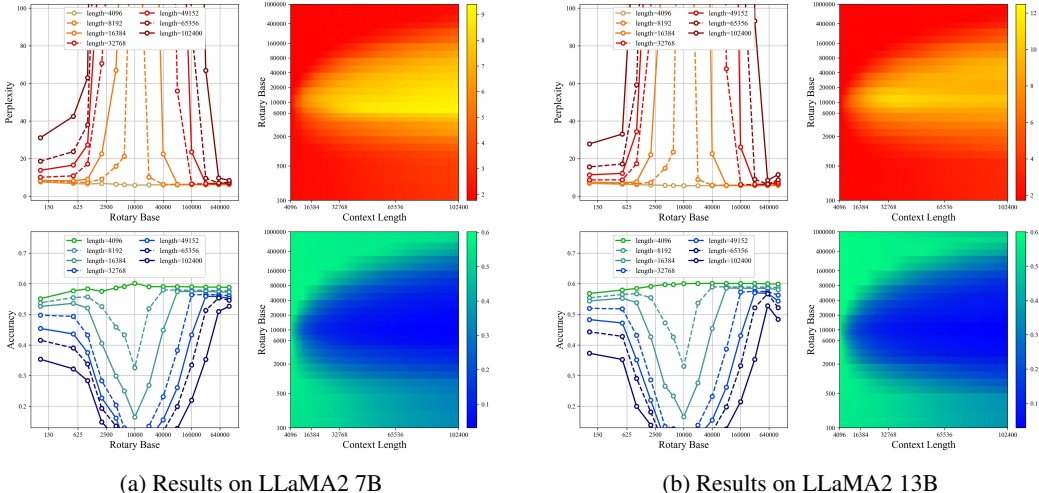

(a) Results on LLaMA2 7B                    (b) Results on LLaMA2 13B

Figure 4: Perplexity and accuracy on the validation data from Books3 dataset (Presser, 2020) of LLaMA2 7B and 13B (Touvron et al., 2023b). In (a) and (b), the first row shows perplexity while the second row shows accuracy, both measured cumulatively. The first column shows the change of extrapolation performance w.r.t. rotary base at different context windows with a line plot, while the second column visualizes it with a heat map. In the heat map of perplexity, the value is log-scaled.

## 3    EXPLANATION

### 3.1    SCALING LAW FOR SMALLER BASES

To understand how RoPE with smaller bases achieves impressive extrapolation within a constrained tuning context, it is crucial to explore the impact of reducing the base. As outlined in Appendix A, a smaller base amplifies the rotary angle $\theta_n = 10000^{-2n/d}$. This shortens $T_n$, the periods of $\sin{(t-s)\theta_n}$ or $\cos{(t-s)\theta_n}$ that RoPE uses to represent relative positions. Figure 5 shows the cosine waves for different dimensions to represent position information. In Figure 5a, it is evident that for smaller bases like 500, any period of $\cos{(t-s)\theta_n}$ is confined to 4096, i.e., the training length of LLaMA2 (Touvron et al., 2023b). In contrast, larger bases like 10000, extend the periods for several dimensions beyond the training length, as detailed in Section 3.2.

Hence, smaller bases lead to a broader range of $\cos$ or $\sin$ inputs during pre-training or fine-tuning, which ensures every dimension of $\boldsymbol{q}_t$ and $\boldsymbol{k}_s$ gets a well-trained representation. Additionally, as the base decreases, three pivotal points emerge, $\pi/2$, $\pi$, and $2\pi$. Only when the $\cos$ or $\sin$ inputs in every dimension span from 0 to $\pi/2$ during training does the RoPE-based LLM recognize the negative values of $\cos$ and the non-monotonicity nature of $\sin$. Similarly, the LLM becomes aware of the non-monotonicity pf $\cos$ and negative $\sin$ values once inputs hit $\pi$. The RoPE-based LLM fully grasps the entire range of $\cos$ and $\sin$ only when the inputs surpass $2\pi$, potentially embracing the periodicity of position embedding in every dimension. Then we propose Theorem 1. as follows.

**Theorem 1.** *(Scaling Law of Smaller Bases)*    For RoPE-based LLMs pre-trained with context length $T_{\text{train}}$, if we adjust the base to $\beta < 10000$ and conduct fine-tuning still with context length $T_{\text{train}}$, the extrapolation performance of RoPE-based LLMs will get improved. If $\beta$ is reduced to $\beta_1, \beta_2, \beta_3$ as calculated below, the $\cos$ or $\sin$ inputs in every dimension will span from 0 to $\pi/2, \pi, 2\pi$ respectively, resulting in a more significant improvement.

$$\beta_1 = \frac{2T_{\text{train}}}{\pi}, \quad \beta_2 = \frac{T_{\text{train}}}{\pi}, \quad \beta_3 = \frac{T_{\text{train}}}{2\pi} \tag{1}$$

Particularly, for LLaMA2(Touvron et al., 2023b), where the context length $T_{\text{train}}$ is 4096, we have $\beta_1 = 2608, \beta_2 = 1304, \beta_3 = 652$. It is worth noting that these three bases align with the pivotal points where improvement speeds up, as previously discussed during base reduction. Theorem 1 tells the improving paradigm of base reduction. Since it does not set an explicit extrapolation upper bound, RoPE with much smaller bases can potentially realize extrapolation to infinite context.

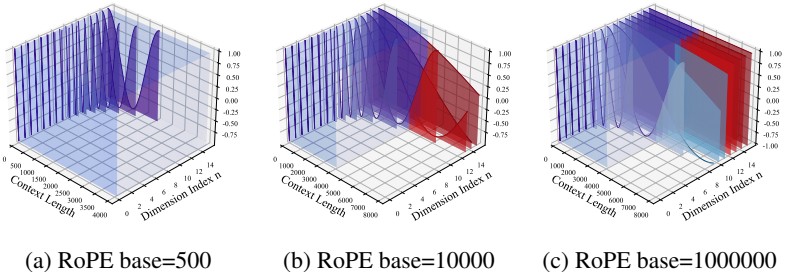

(a) RoPE base=500       (b) RoPE base=10000       (c) RoPE base=1000000

Figure 5: The visualized relationship among the period, training Length, and extrapolation. Consider a RoPE-based LLM with a head dimension size of 32, namely 16 rotary angles $\theta_n$ across various dimensions. Within each illustration, we visually represent the periods of $\cos(t-s)\theta_n$ for these dimensions using parallel purple planes. These are contrasted against the 4096 training context length, shown as a deep blue box. **(a)** For RoPE tuned with base 500, all periods of $\cos(t-s)\theta_n$ are limited within the training context. **(b)** For vanilla RoPE with base 10000, the periods of dimensions past the critical dimension (in red) stretch beyond the training context. **(c)** For RoPE tuned with base 1000000, although some updated periods colored (in sky blue) might surpass the training context, only dimensions past the critical dimension fail to extrapolate.

## 3.2 CRITICAL DIMENSION FOR EXTRAPOLATION

In comparison to much smaller bases like 500, where each period of $\cos(t-s)\theta_n$ fits within the training context, the default base in RoPE (Su et al., 2021), which is 10000, causes periods of certain dimensions to extend beyond the training context, as visualized in Figure 5b. Therefore, for RoPE-based LLMs, there exists a specific feature dimension, $d_{\text{extra}}$. For dimensions before $d_{\text{extra}}$, the periods of corresponding $\theta_n$ remain shorter than $T_{\text{train}}$, while for those after $d_{\text{extra}}$, the periods stretch beyond $T_{\text{train}}$. In other words, essentially, $d_{\text{extra}}$ is the number of dimensions where $\cos(t-s)\theta_n$ and $\sin(t-s)\theta_n$ can cycle through their values within one period during pre-training or fine-tuning.

Consequently, for dimensions beyond $d_{\text{extra}}$, when RoPE-based LLMs extrapolate beyond $T_{\text{train}}$, the absolute position information of newly added tokens and the relative positional information in relation to previous tokens become out-of-distribution (OOD). This misalignment means the attention scores related to these dimensions, as illustrated in Equation 2, deviate from their expected distribution, causing a noticeable out-of-distribution in overall attention scores, thus leading to the extrapolation issue. We refer to this key dimension as the ***Critical Dimension for RoPE-based extrapolation***, which is formally defined and calculated as shown in Lemma 1.

$$
\begin{aligned}
\boldsymbol{A}_{t,s} &= \operatorname{Re}\left[\underbrace{\sum_{n=0}^{d/2-1} \tilde{q}_t^{(n)} \tilde{k}_s^{(n)*} e^{i(t-s)\theta_n}}_{\text{full attention scores in RoPE}}\right] \\
&= \operatorname{Re}\left[\underbrace{\sum_{n=0}^{d_{\text{extra}}/2-1} \tilde{q}_t^{(n)} \tilde{k}_s^{(n)*} e^{i(t-s)\theta_n}}_{\text{reliable part for extrapolation}} + \underbrace{\sum_{n=d_{\text{extra}}/2}^{d/2-1} \tilde{q}_t^{(n)} \tilde{k}_s^{(n)*} e^{i(t-s)\theta_n}}_{\text{OOD part for extrapolation}}\right].
\end{aligned} \tag{2}
$$

**Lemma 1.** *(Definition of Critical Dimension)* For RoPE-based LLMs pre-trained with context length $T_{\text{train}}$, assuming that the size of self-attention head is $d$, there are at most the preceding $d_{\text{extra}}$ dimensions that perceive complete periodic information thus receiving sufficient training for extrapolation, which is formally described as follows:

$$
\begin{aligned}
T_n &= \frac{2\pi}{\theta_n} = 2\pi \cdot 10000^{\frac{2n}{d}} \leq T_{\text{train}}, \quad \text{for } n = 0, \cdots, d_{\text{extra}}/2 - 1, \\
T_n &= \frac{2\pi}{\theta_n} = 2\pi \cdot 10000^{\frac{2n}{d}} > T_{\text{train}}, \quad \text{for } n = d_{\text{extra}}/2, \cdots, d/2 - 1.
\end{aligned} \tag{3}
$$

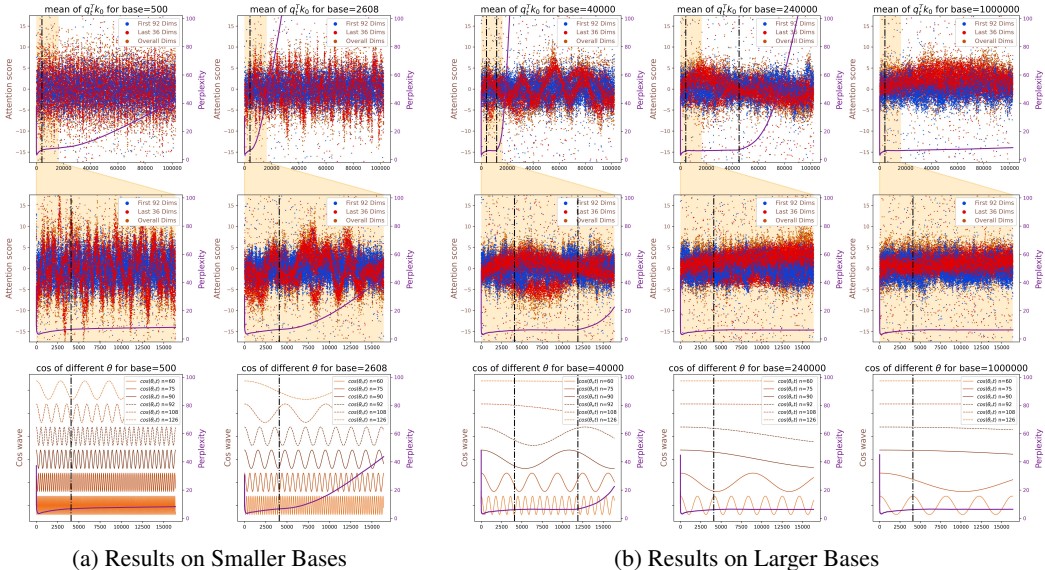

(a) Results on Smaller Bases                    (b) Results on Larger Bases

Figure 6: The relation between attention scores in first 92 and last 36 dimensions with the extrapolation performance in LLaMA 7B (Touvron et al., 2023b) fine-tuned with larger or smaller bases. The first row shows how average attention score in the first 92 and last 36 dimensions and the perplexity changes in 100K context length. The second row highlights the changes in the first 16K tokens. The third row visualizes that the period of critical dimension determines the attention score explosion and extrapolation issue. The black lines stand for training length or max context size.

Then we define $d_{\text{extra}}$ as the critical dimension for RoPE-based extrapolation and calculate it given

$$d_{\text{extra}} = 2 \left\lceil \frac{d}{2} \log_{10000} \frac{T_{\text{train}}}{2\pi} \right\rceil. \tag{4}$$

For LLaMA2(Touvron et al., 2023b), the critical dimension $d_{\text{extra}}$ is 92. This implies that only the first 92 dimensions of the $\boldsymbol{q}_t, \boldsymbol{k}_s$ vectors of LLaMA2 have seen the complete positional information during the pre-training phase and are adequately trained. In other words, the last 36 dimensions lack sufficient training, contributing to the extrapolation challenges seen in RoPE-based LLMs (Chen et al., 2023; Han et al., 2023). The critical dimension plays a key role in enhancing extrapolation. A further discussion of the critical dimension is presented in Section 3.4. Here, we examine the attention score changes in the initial 92 versus the final 36 dimensions in relation to relative positions when the base is reduced. As is shown in Figure 6a, the attention scores of RoPE with smaller bases have effectively captured the oscillations from sin and cos within training, mitigating OOD concerns during extrapolation. Moreover, as the base becomes smaller, the perception becomes more comprehensive, resulting in improved extrapolation performance.

### 3.3 SCALING LAW FOR LARGER BASES

Based on the concept of the critical dimension, we can clarify the extrapolation results when fine-tuning RoPE with larger bases at the original context length. For LLaMA2 (Touvron et al., 2023b), since the periods of the first 92 dimensions fit within the training length, these feature dimensions start with a strong foundation for fine-tuning, adjusting to the new periodic shifts of positional embedding for extended contexts. Therefore, when RoPE is fine-tuned with a larger base value like 1000000, even though the tuning length is shorter than the extended periods corresponding to larger bases, these dimensions can still represent positional information correctly as is shown in Figure 5c.

However, for the last 36 dimensions, the absence of a full understanding of periodicity leads to overfitting. Furthermore, when the base is expanded, extending the period, these dimensions still fail to capture the entire positional information within the context length. So these dimensions are reliable only when the value of $\theta_n(t - s)$ is previously observed. Therefore, we can use the updated period

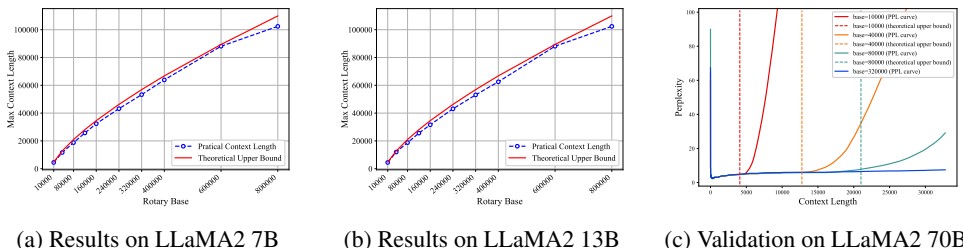

| (a) Results on LLaMA2 7B | (b) Results on LLaMA2 13B | (c) Validation on LLaMA2 70B |

Figure 7: The comparison in 100K context length and 7B/13B model between max practical context length and extrapolation upper bound predicted by Theorem 2. for RoPE tuned with larger bases on pre-training context. We also validate our theory in 32K context length and 70B size and find that the tuning points of the perplexity curve match the theoretical upper bounds of length extrapolation.

of the critical dimension as an upper bound for extrapolation in RoPE-based LLM. As a result, we obtain Theorem 2., the scaling law for RoPE-based extrapolation with larger bases, which tells the relation between base value and extrapolation upper bound and thus answers Q2 in the Introduction.

**Theorem 2.** *(Scaling Law of Larger Bases)* For RoPE-based LLMs pre-trained with context length $T_{\text{train}}$, if we adjust the base to $\beta > 10000$ and conduct fine-tuning still with context length $T_{\text{train}}$, the extrapolation performance of RoPE-based LLMs will get improved. The extrapolation upper bound of RoPE-based LLM with larger bases, $T_{\text{extra}}$, is calculated as follows:

$$T_{\text{extra}} = 2\pi \cdot \beta^{d_{\text{extra}} \cdot \frac{1}{d}} = 2\pi \cdot \beta^{\left\lceil \frac{d}{2} \log_{10000} \frac{T_{\text{train}}}{2\pi} \right\rceil \cdot \frac{2}{d}}. \tag{5}$$

Inversely, if there is an expected extrapolation upper bound $\tilde{T}_{\text{extra}}$, then the smallest capable base $\beta_0$ is calculated as follows.

$$\beta_0 = 10000^{\log_{\frac{T_{\text{train}}}{2\pi}} \frac{\tilde{T}_{\text{extra}}}{2\pi}}. \tag{6}$$

For extrapolation within a limited context, we can derive the suggested $\beta_0$ based on the expected context length. $\beta_0$ in Equation 6 is referred to as ***critical base*** for extrapolation and discussed in detail in Appendix C. To support the claims of Theorem 2, as illustrated in Figure 6b, we examined the attention scores of the first 92 and final 36 dimensions under various larger bases. Notably, while the attention scores of the first 92 dimensions remain relatively stable regardless of relative positions, the last 36 dimensions show significant variations. Reviewing the perplexity increase curves, it is evident that once the context length surpasses the extrapolation upper bound, the last 36 dimensions encounter unfamiliar positional information, leading to OOD attention scores and a sharp rise in perplexity. To further validate Theorem 2, we compare the max supported context lengths for different bases with the extrapolation upper bound derived from Theorem 2 in Figure 7. Impressively, there is a remarkable alignment between empirical results and theoretical predictions.

### 3.4 Further Validation for Extrapolation

To further illustrate the causal relationship between critical dimension and extrapolation, we undertake the following three experiments. First, we set a max index of 4096 for the position embedding of the last 36 dimensions of $q_t, k_s$ in LLaMA2 7B (Touvron et al., 2023b). We find that apart from an outlier in one attention head, the attention score variations are considerably reduced, leading to better extrapolation. Second, we visualize the attention score of the first 92 and the last 36 dimensions during extrapolation with Dynamic NTK (bloc97, 2023a) in Figure 8a. Consistent with our theory, the attention scores of the last 36 dimensions display enhanced consistency compared with straightforward extrapolation. Lastly, we remove the final 36 dimensions of $q_t, k_s$ in LLaMA2 7B (Touvron et al., 2023b) and fine-tuned using the setup in Appendix B.1. Remarkably, post-trimming fine-tuning substantially surpassed direct fine-tuning, allowing extrapolation beyond 20K tokens, as illustrated in Figure 8b. This offers compelling evidence for the correlation between the critical dimension, attention score shifts, and extrapolation upper bound. It confirms that interpreting and enhancing RoPE-based LLM extrapolation from a periodic viewpoint is both valid and effective.

|  | 128K | 256K | 512K | 1M |
|---|---|---|---|---|
| base=500 | 9.49 | 12.41 | 23.99 | 51.28 |
| base=500 log-scaled | 9.13 | 10.01 | 12.07 | 19.07 |
| base=1000000 | 7.07 | 76.82 | 1755.87 | 4914.81 |

Table 1: Perplexity on Books3 of LLaMA2 7B with base 500 and 1000000 tuned on 16K context.

In summary, we establish a comprehensive framework from a periodic perspective and identify the fundamental factors, the critical dimension for RoPE-based extrapolation. This not only unveils the root causes of attention score explosion highlighted in Chen et al. (2023) and Han et al. (2023), but also intuitively demonstrates how adjusting the base during fine-tuning can elevate the extrapolation capabilities of RoPE-based LLMs. Furthermore, we answer Q3 in the Introduction and introduce the ***Extended Scaling Law of RoPE-based Extrapolation*** in Appendix C, the combination of Theorem 1., Lemma 1. and Theorem 2., for tuning in a longer context. Additionally, we also present the instructive value of our theory for other works focused on achieving longer context in the testing phase in Appendix D. Finally, as depicted in Figure 1, tuning RoPE with a base of 500 or 1000000 can both outperform Linear PI (Chen et al., 2023) and NTK method (bloc97, 2023b;a). Besides, we compare the performance of LLaMA2 7B Touvron et al. (2023b) with base 500 and 1000000 in 1M context length in Table 1 and find that for extrapolating to unpredictable length, RoPE with base 500, has remarkable advantages, especially when combined with log-scaled attention (Su, 2023b).

## 4 RELATED WORK

Long sequence modeling is an important issue in LLM applications. It faces the following three challenges: the quadratic computational complexity, the difficulty of parallelism for sequence length, and the collapse of the performance beyond the training length. First of all, due to the quadratic complexity of the self-attention mechanism in Transformers, long sequence modeling requires a large computational overhead. Early efficient Transformers reduced computational overhead through methods such as recalculation like ReFormer(Kitaev et al., 2020) and sparse attention such as LongFormerBeltagy et al. (2020) and BigBird(Zaheer et al., 2020). After that, Lin-

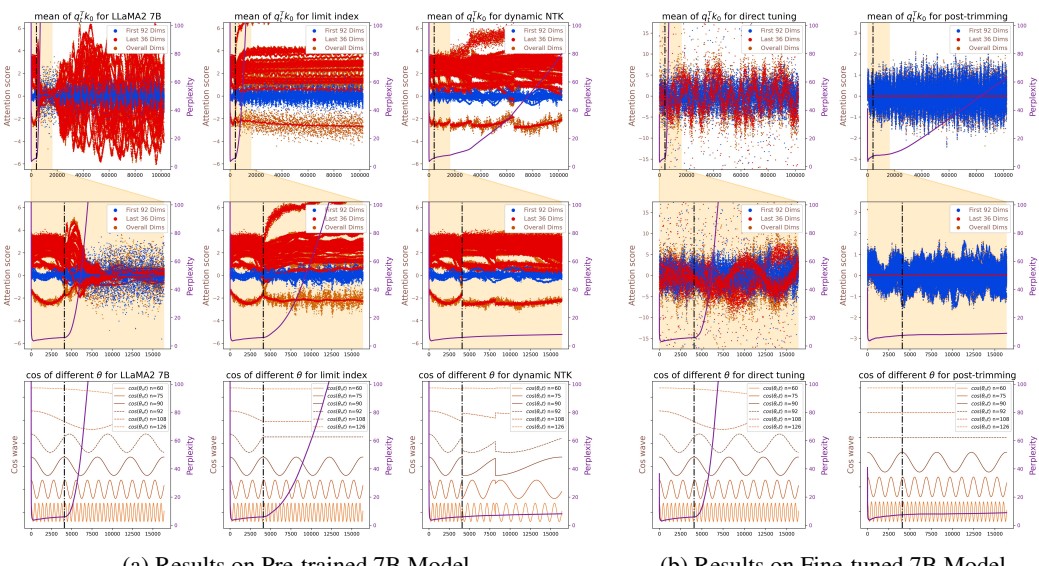

(a) Results on Pre-trained 7B Model    (b) Results on Fine-tuned 7B Model

Figure 8: The relation between attention scores in first 92 and last 36 dimensions with the extrapolation performance in LLaMA 7B (Touvron et al., 2023b) evaluated or fine-tuned with original bases 10000. The meaning of each row is the same as that in Figure 6.

Former(Wang et al., 2020) linearizes the self-attention complexity through low-rank approximation, and Performer(Choromanski et al., 2021) expands the softmax calculation through a kernel function. Compared with these efficient attention work based on approximation, FlashAttention(Dao et al., 2022; Dao, 2023) reduces HBM reading and writing and turns self-attention into a form that can be iteratively calculated in a loop, which greatly improves the utilization of computing resources and has become the standard for long-sequence modeling.

The second challenge is parallelism for sequence length, that is, Sequence Parallelism (SP) (Li et al., 2023c). Since the memory and calculation of a single device are limited, ColossalAI-SP (Li et al., 2023c) first proposes the segmentation and parallelism for the sequence dimension in addition to tensor parallelism for the hidden dimension and pipeline parallelism for model depth. On this basis, Ring Attention(Liu et al., 2023) uses blockwise self-attention to split long sequences into different devices and overlap the communication of key-value blocks. Besides, LightSeq(Li et al., 2023b) further improves the efficiency of long sequence modeling through load balancing for causal language modelings and a re-materialization-aware checkpointing strategy. Although the above sequence parallelism can achieve infinitely long sequence modeling, they ignore the compatibility with the existing efficient self-attention mechanisms such as FlashAttention(Dao et al., 2022; Dao, 2023) to achieve input with almost infinity. On the contrary, Megatron-SP(Korthikanti et al., 2022) only uses sequence parallelism during Dropout and Layernorm operations, thereby reducing activation redundancy. In addition, DeepSpeed-Ulysses(Jacobs et al., 2023) uses an alternative all-to-all collective communication gathering weight for attention computation when segmenting the sequence, avoiding communication overhead that originally increases with length.

Recently, there have been a great deal of efforts devoted to solving the collapse of the performance beyond the training length and expanding the context length of RoPE-based LLMs to 16K, 32K, or even 100K (Chen et al., 2023; Su, 2023c; bloc97, 2023b). Considering that LLMs, such as LLaMA2 (Touvron et al., 2023b), have already acquired sufficient knowledge in the pre-training stage and demonstrated excellence in short-context tasks, the emerging extrapolation improvements have primarily focused on the fine-tuning and the testing phase. Initially, the context window is a well-discussed strategy for any LLM without further training. Effective as it is, it prohibits tokens of their deserved global attention (Han et al., 2023) or destroys the order between token chunks (Ratner et al., 2022; Su, 2023a), which is unsupported for long document summarization (Su, 2023a). Contrastively, Chen et al. (2023) extends the context window to 16K by linearly reducing the rotary angles with $\lambda = T_{\text{extra}}/T_{\text{train}}$ to align the input position index within the original context size. Besides, it is worth noting the remarkable effect of the Neural Tangents Kernel (NTK) method (bloc97, 2023b), especially the dynamic version (bloc97, 2023a) [2] simply decreases the rotary angle exponentially with scaling coefficient $\alpha$ as a function w.r.t. the inference length, which is discussed in detail in Appendix D. Dynamic NTK enables LLaMA2(Touvron et al., 2023b) to extrapolate directly and keeps good performance within 16K context length. However, Dynamic NTK still presents an obvious degradation in performance around 32K context size as shown in Figure 1. Inspired by NTK methods, Rozière et al. (2023) achieves 100K context window for code by increasing the rotary base to 1000000 and further training with 16K context length. Besides, Pal et al. (2023) also extends the context window to 32K by a new truncation strategy for refining the down-sampling method, which even sets some rotary angles to zero. Additionally, Peng et al. (2023) proposes a scaling function, fine-tunes LLaMA2 with 64K context, and finally achieves the 128K context length. Effective as they are, these methods still face a disastrous collapse after an unpredictable extrapolation bound.

## 5 CONCLUSION

In summary, we initially highlight an intriguing observation: fine-tuning RoPE (Su et al., 2021) with either a larger or smaller base using the original pre-training context can boost the length extrapolation of RoPE-based LLMs. We then elucidate this observation through a unified theoretical lens rooted in a periodic view, *Scaling Laws of RoPE-based Extrapolation*. This framework clarifies the score of the RoPE-based extrapolation challenge and offers insights into how base modifications in fine-tuning or inference can enhance RoPE-based extrapolation. Finally, we provide the strategy for both extrapolating LLaMA2 (Touvron et al., 2023b) to a limited context and unpredictable inputs.

---

[2]In this work, we follow the form of scaling function in QWenLM (Alibaba, 2023)

ACKNOWLEDGEMENTS

This work is supported by Shanghai Artificial Intelligence Laboratory. Special thanks to Qipeng Guo, Shuo Zhang, and Kai Lv, whose invaluable suggestions have greatly contributed to this work.

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

## A PRELIMINARY

### A.1 RoPE FROM SEQUENCE DOMAIN

Transformer models require the integration of explicit positional information through positional embeddings to effectively discern the order of input sequences (Vaswani et al., 2017). In this work, we direct our attention to the specific instance of positional encoding known as Rotary Position Embedding (RoPE) (Su et al., 2021), as prominently featured in the architecture of the LLaMA model (Touvron et al., 2023a;b). Given a query vector $\boldsymbol{q}_t = \left[q_t^{(0)}, \cdots, q_t^{(d-1)}\right] \in \mathbb{R}^d$ at position $t$ and a key vector $\boldsymbol{k}_s = \left[k_s^{(0)}, \cdots, k_s^{(d-1)}\right] \in \mathbb{R}^d$ at position $s$, RoPE first splits $q_t, k_s$ into pairs on the direction of feature dimensions, with every two dimensions forming a complex number, or a vector in the complex plane as follows:

$$
\begin{aligned}
\tilde{\boldsymbol{q}}_t &= \left[\tilde{q}_t^{(0)}, \cdots, \tilde{q}_t^{(d/2-1)}\right] & \tilde{q}_t^{(n)} &= q_t^{(2n)} + i q_t^{(2n+1)} \\
\tilde{\boldsymbol{k}}_s &= \left[\tilde{k}_s^{(0)}, \cdots, \tilde{k}_s^{(d/2-1)}\right] & \tilde{k}_s^{(n)} &= k_s^{(2n)} + i k_s^{(2n+1)}
\end{aligned}
\tag{7}
$$

After that, RoPE injects the position information by an element-wise multiplication between the pre-processed $\tilde{\boldsymbol{q}}_t, \tilde{\boldsymbol{k}}_s$ and a list of $\boldsymbol{\theta}$-parameterized rotary vectors in the complex plane. When attention

is calculated, relative position information $t - s$ is acquired through $\cos$ and $\sin$.

$$
\begin{aligned}
\boldsymbol{A}_{t,s} &= \mathrm{Re}\left[\left(\tilde{\boldsymbol{q}}_t \odot e^{it\boldsymbol{\theta}}\right) \cdot \left(\tilde{\boldsymbol{k}}_s \odot e^{is\boldsymbol{\theta}}\right)^T\right] \\
&= \mathrm{Re}\left[\sum_{n=0}^{d/2-1} \tilde{q}_t^{(n)} e^{it\theta_n}\left(\tilde{k}_s^{(n)} e^{is\theta_n}\right)^*\right] = \mathrm{Re}\left[\sum_{n=0}^{d/2-1} \tilde{q}_t^{(n)} \tilde{k}_s^{(n)*} e^{i(t-s)\theta_n}\right]. \\
&= \sum_{n=0}^{d/2-1} \left(q_t^{(2n)} k_s^{(2n)} + q_t^{(2n+1)} k_s^{(2n+1)}\right) \cos{(t-s)\theta_n} + \\
&\quad\ \left(q_t^{(2n)} k_s^{(2n+1)} - q_t^{(2n+1)} k_s^{(2n)}\right) \sin{(t-s)\theta_n}
\end{aligned}
\tag{8}
$$

While RoPE can theoretically convey the relative information at any context length, RoPE still fails to extrapolate practically. It is worth noting that rotary angles $\boldsymbol{\theta}$ in Equation 8 play an important role. In the vanilla design of RoPE, $\boldsymbol{\theta}$ is defined as Equation 9. Different angles correspond to different features and that is the starting point of most RoPE-based extrapolation methods (bloc97, 2023b; Rozière et al., 2023; Pal et al., 2023) shown in Table 2.

$$
\boldsymbol{\theta} = \left[\theta_0, \cdots, \theta_{d/2-1}\right] \quad \theta_n = 10000^{-2n/d}.
\tag{9}
$$

For example, the NTK method (bloc97, 2023b;a) reduces the rotary angle under the maintenance of its value form, thus enabling RoPE-based LLM to adapt to position information in a longer context without further training.

## A.2 RoPE from Frequency Domain

From a frequency domain perspective, the rotation operation of RoPE can be viewed as the Inverse Fourier Transform from the frequency domain to the time domain.

$$
\begin{aligned}
\boldsymbol{A}_{t,s} &= \mathrm{Re}\left[\sum_{n=0}^{d/2-1} \tilde{q}_t^{(n)} \tilde{k}_s^{(n)*} e^{i(t-s)\theta_n}\right]. \\
&= \mathrm{Re}\left[\mathcal{F}_{\boldsymbol{\theta}}^{-1}\left[\tilde{\boldsymbol{q}}_t^{(n)} \tilde{\boldsymbol{k}}_s^{(n)*}\right]\right]
\end{aligned}
\tag{10}
$$

At this juncture, the learning objective for RoPE-based LLM is essentially to understand features in terms of frequency. Depending on the value of $\theta_n$, the higher dimensions correspond to the longer period as shown in Equation 11 as well as the lower-frequency features reflecting the longer contextual semantic relationship (Chen et al., 2023; Rozière et al., 2023).

$$
T_n = \frac{2\pi}{\theta_n} = 2\pi \cdot 10000^{2n/d}, \quad \text{for } n = 0 \cdots \frac{d}{2} - 1.
\tag{11}
$$

However, as illustrated in Figure 5, the trigonometric functions of lower-frequency features do not complete a full period within the training context. As a result, RoPE-based LLMs might not fully recognize the periodic nature of $\sin$ and $\cos$ waves, leading to inadequate training. Consequently, these lower-frequency features are more susceptible to under-fitting or over-fitting. Hence, the number of well-trained dimensions is essential and this is the critical dimension raised in Section 3.2.

## B Appendix

### B.1 Experiment Setup

We conduct experiments on the pre-trained 7B and 13B LLaMA2 models (Touvron et al., 2023b). For fine-tuning 7B and 13B models, we use 32 A100 GPUs and adopt ZeRO3 strategies (Rajbhandari et al., 2020). We use AdamW (Loshchilov & Hutter, 2019) with $\beta_1 = 0.9$ and $\beta_2 = 0.999$. We set the learning rate to $2 \times 10^{-5}$ with no warmup. We set the max gradient norm to 2.5 for 7B and 1 for 13B respectively. We set the weight decay to zero.

| Name | Applying Phase | Training Length | Context Window | Ratio | Auxiliary Window | Note |
|------|----------------|-----------------|----------------|-------|------------------|------|
| xPos (Sun et al., 2023) | Pre-training | 1K | 4K | ×4 | ✔ | BCA |
| Linear PI (Chen et al., 2023) | Fine-tuning | 16K | 16K | ×1 | ✗ | |
| Giraffe (Pal et al., 2023) | Fine-tuning | 4K | 32K | ×8 | ✗ | truncated |
| Code LLaMA (Rozière et al., 2023) | Fine-tuning | 16K | 100K+ | ×6 | ✗ | |
| YaRN (Peng et al., 2023) | Fine-tuning | 64K | 128K | ×2 | ✔ | window ppl |
| Fixed NTK (bloc97, 2023b) | Inference | 4K | 16K | ×4 | ✗ | |
| Dynamic NTK (bloc97, 2023a) | Inference | 4K | 16K+ | ×4 | ✗ | |
| PCW (Ratner et al., 2022) | Inference | 2K | 6K | ×3 | ✔ | for ICL |
| NBCE (Su, 2023a) | Inference | 2K | 10K | ×5 | ✔ | for ICL |
| LM-Infinite (Han et al., 2023) | Inference | 4K | 32K+ | ×8 | ✔ | Λ-shaped |
| Scaling RoPE (Ours) | Fine-tuning | 4K | 100K+ | ×25 | ✗ | base=1000000 |
| | | 16K | **1M+** | **×64** | | base=500 |

Table 2: RoPE-based Extrapolation Strategies. For those applied in fine-tuning, the max training length is the tuning context length. For PCW and NBCE, the context size is the max training length times the optimal chunk number. It is worth noting that Scaling RoPE, namely tuning RoPE with smaller or larger bases discussed in this work, realizes the max context window and extension ratio.

For fine-tuning RoPE with different bases, we set the global batch size to 128, tuning the context length to 4K, the same as the training length, and the evaluating context length to 100K. We fine-tune the models for 1K steps using the next token prediction objective with training data from the Pile (Gao et al., 2021) and compare the tuning performance on the validation set of Books3 subset (Presser, 2020) from the Pile. We fine-tune LLaMA2 with CoLLiE(Lv et al., 2023), a collaborative toolbox for tuning large language models in an efficient way, and conduct evaluation discussed below with OpenCompass[3]. Both training and testing are accelerated by FlashAttention-2 (Dao, 2023).

We compare results with mainstream extrapolating strategies, such as Linear Position Interpolation (Chen et al., 2023) and NTK method (bloc97, 2023b;a). For fine-tuning with Linear PI, we set the global batch size to 64 and tuning length to 8K, which follows Chen et al. (2023). For fine-tuning with 16K context length, wet set the global batch size 32. When we fine-tune LLaMA2 7B with the last 36 dimensions being cut off, discussed in Section 3.4, we set the softmax scale $\sqrt{92}$, the square root of updated dimension size of $q_t, k_s$, and keep other setups the same. Except for the position embedding, we do not modify the architecture of LLaMA2 (Touvron et al., 2023b).

### B.2 SHORT CONTEXT VALIDATION

Next, we validate whether Scaling RoPE, namely the fine-tuning RoPE with different bases on original context length, has a side effect on LLM. We use short context tasks including those listed in the Hugging Face Open LLM Leaderboard(Face, 2023), such as 0-shot Lambada(Paperno et al., 2016), Hellaswag(Zellers et al., 2019), NQ(Kwiatkowski et al., 2019), TriviaQA(Joshi et al., 2017), OBQA(Mihaylov et al., 2018), PIQA(Bisk et al., 2020), TruthfulQA(Lin et al., 2022), Winogrande(Sakaguchi et al., 2020), SuperGLUE(Wang et al., 2019), ARC-easy/challenge(Clark et al., 2018) and 5-shot MMLU(Hendrycks et al., 2021), to assess the performance of RoPE-based LLM after the fine-tuning process. The results obtained are shown in the Table 3 and Table 4. The results

---

[3]https://opencompass.org.cn/

|  | Lambada | SuperGLUE | Hellaswag | PIQA | Winogrande | OBQA |
|---|---|---|---|---|---|---|
| LLaMA2 7B | 73.30 | 50.43 | 69.45 | 76.66 | 61.25 | 32.00 |
| NTK fixed $\alpha = 8$ | 67.53 | 51.13 | 68.43 | 76.28 | 61.48 | 29.40 |
| Linear PI $\lambda = 2$ | 72.64 | 50.71 | 69.75 | 76.01 | 61.56 | 40.60 |
| Linear PI $\lambda = 4$ | 71.82 | 52.34 | 69.48 | 76.33 | 61.56 | 30.40 |
| base=500 | 69.51 | 48.34 | 66.46 | 74.97 | 59.67 | 31.00 |
| base=652 | 68.78 | 46.86 | 64.00 | 75.35 | 60.22 | 28.80 |
| base=1304 | 69.24 | 47.22 | 65.51 | 76.06 | 60.30 | 25.00 |
| base=2608 | 69.38 | 48.04 | 65.61 | 75.24 | 60.93 | 28.00 |
| base=10000 | 72.62 | 51.49 | 69.60 | 75.95 | 61.40 | 38.00 |
| base=40000 | 72.77 | 51.02 | 69.46 | 76.06 | 61.33 | 36.20 |
| base=80000 | 72.68 | 51.20 | 69.44 | 76.01 | 60.69 | 37.40 |
| base=160000 | 72.54 | 50.24 | 69.07 | 76.01 | 60.77 | 36.80 |
| base=400000 | 72.25 | 51.08 | 68.86 | 75.68 | 60.85 | 36.00 |
| base=600000 | 72.09 | 51.02 | 68.90 | 75.63 | 61.33 | 34.80 |
| base=1000000 | 71.80 | 50.84 | 68.82 | 75.63 | 60.77 | 35.40 |
| base=500, 16K | 68.62 | 48.20 | 62.35 | 73.67 | 58.33 | 25.00 |
| base=10000, 16K | 72.17 | 50.60 | 67.68 | 75.79 | 60.22 | 28.80 |
| base=40000, 16K | 70.77 | 48.03 | 67.69 | 76.33 | 61.33 | 28.20 |
| base=80000, 16K | 72.48 | 51.29 | 69.54 | 76.33 | 61.09 | 32.60 |
| base=120000, 16K | 72.66 | 50.34 | 69.54 | 76.22 | 61.96 | 32.20 |
| base=1000000, 16K | 71.14 | 49.80 | 68.45 | 75.90 | 61.88 | 29.80 |
| LLaMA2 13B | 76.48 | 52.35 | 74.22 | 79.54 | 63.85 | 42.60 |
| NTK fixed $\alpha = 8$ | 72.56 | 52.64 | 73.56 | 79.71 | 62.67 | 37.00 |
| Linear PI $\lambda = 2$ | 75.18 | 63.15 | 74.93 | 78.94 | 63.69 | 51.40 |
| Linear PI $\lambda = 4$ | 74.29 | 61.64 | 74.81 | 79.38 | 63.77 | 50.60 |
| base=500 | 72.56 | 52.81 | 71.38 | 76.88 | 64.17 | 35.20 |
| base=652 | 73.78 | 52.54 | 71.62 | 77.48 | 62.83 | 35.20 |
| base=1304 | 74.21 | 52.19 | 72.38 | 77.37 | 64.01 | 36.80 |
| base=2608 | 74.54 | 52.54 | 73.63 | 78.02 | 63.22 | 43.40 |
| base=10000 | 75.12 | 53.19 | 75.19 | 79.00 | 62.98 | 55.00 |
| base=40000 | 74.87 | 53.11 | 74.76 | 79.60 | 62.98 | 52.60 |
| base=80000 | 74.89 | 52.37 | 74.76 | 79.27 | 62.90 | 53.80 |
| base=160000 | 74.89 | 52.63 | 74.50 | 79.05 | 62.75 | 54.20 |
| base=400000 | 74.23 | 52.47 | 74.50 | 78.89 | 62.19 | 54.40 |
| base=600000 | 74.19 | 54.14 | 74.24 | 79.00 | 63.77 | 53.60 |
| base=1000000 | 74.58 | 50.82 | 74.36 | 78.67 | 62.98 | 51.80 |
| base=500, 16K | 69.34 | 49.28 | 67.23 | 76.50 | 58.48 | 29.60 |
| base=10000, 16K | 74.00 | 53.44 | 73.16 | 78.89 | 63.93 | 35.00 |
| base=40000, 16K | 75.06 | 54.08 | 74.33 | 78.78 | 62.51 | 38.00 |
| base=80000, 16K | 74.77 | 53.28 | 75.11 | 79.05 | 63.61 | 38.20 |
| base=120000, 16K | 75.06 | 53.91 | 74.85 | 78.94 | 63.61 | 43.60 |
| base=1000000, 16K | 74.71 | 51.95 | 74.68 | 79.11 | 63.30 | 40.80 |

Table 3: Short context validation results of scaling RoPE with different base and other extrapolating methods in LLaMA2 (Touvron et al., 2023b). The first box is the result of 7B size and the second box is those of 13B version. LLaMA2 7B or 13B corresponds to the original result as well as that of Dynamic NTK (bloc97, 2023a) since Dynamic NTK does not affect the model within the training context. The final score of SuperGLUE is the average score of all corresponding subtasks.

of Scaling-RoPE are comparable with that of the pre-trained or directly fine-tuned LLaMA2 (Touvron et al., 2023b). This indicates our approach not only enhances the extrapolation performance but also maintains the inherent knowledge of LLM.

| | NQ | TriviaQA | TruthfulQA | ARC-e | ARC-c | MMLU |
|---|---|---|---|---|---|---|
| LLaMA2 7B | 19.06 | 54.09 | 35.09 | 50.26 | 36.61 | 46.78 |
| NTK fixed $\alpha = 8$ | 19.89 | 53.16 | 35.09 | 48.68 | 34.58 | 39.36 |
| Linear PI $\lambda = 2$ | 18.64 | 52.57 | 34.80 | 56.44 | 35.93 | 47.27 |
| Linear PI $\lambda = 4$ | 19.78 | 52.23 | 35.09 | 55.91 | 36.95 | 46.80 |
| base=500 | 20.36 | 52.43 | 35.09 | 53.09 | 38.64 | 38.70 |
| base=652 | 17.12 | 48.90 | 34.36 | 46.74 | 33.90 | 33.43 |
| base=1304 | 17.70 | 51.27 | 35.09 | 49.38 | 37.63 | 33.46 |
| base=2608 | 16.01 | 51.21 | 34.06 | 48.15 | 34.58 | 37.07 |
| base=10000 | 20.14 | 55.14 | 35.53 | 54.32 | 38.31 | 47.73 |
| base=40000 | 17.34 | 55.27 | 35.23 | 55.03 | 36.95 | 47.34 |
| base=80000 | 17.81 | 54.72 | 35.23 | 53.09 | 37.63 | 46.87 |
| base=160000 | 18.98 | 55.28 | 35.09 | 53.26 | 38.64 | 46.24 |
| base=400000 | 20.39 | 54.91 | 35.23 | 52.03 | 38.64 | 46.01 |
| base=600000 | 19.20 | 54.75 | 35.67 | 53.26 | 37.97 | 46.39 |
| base=1000000 | 19.50 | 54.76 | 35.82 | 52.20 | 37.29 | 46.29 |
| base=500, 16K | 11.58 | 37.95 | 34.36 | 46.21 | 36.95 | 28.79 |
| base=10000, 16K | 15.29 | 50.31 | 34.50 | 49.91 | 36.61 | 42.71 |
| base=40000, 16K | 15.32 | 49.16 | 34.94 | 49.38 | 31.86 | 45.44 |
| base=80000, 16K | 16.51 | 51.16 | 34.36 | 52.20 | 35.25 | 45.08 |
| base=120000, 16K | 16.15 | 51.84 | 34.21 | 51.85 | 34.24 | 43.75 |
| base=1000000, 16K | 16.68 | 53.14 | 34.80 | 50.79 | 34.92 | 41.83 |
| LLaMA2 13B | 24.82 | 60.69 | 34.65 | 59.44 | 40.00 | 55.77 |
| NTK fixed $\alpha = 8$ | 22.77 | 60.81 | 34.80 | 58.38 | 40.34 | 52.04 |
| Linear PI $\lambda = 2$ | 25.96 | 59.21 | 34.94 | 63.49 | 40.00 | 57.20 |
| Linear PI $\lambda = 4$ | 26.07 | 59.34 | 35.67 | 64.20 | 39.66 | 56.08 |
| base=500 | 11.50 | 57.54 | 33.77 | 57.67 | 38.98 | 47.98 |
| base=652 | 15.98 | 58.67 | 35.23 | 58.73 | 38.31 | 50.49 |
| base=1304 | 13.60 | 59.12 | 35.23 | 58.73 | 41.36 | 53.59 |
| base=2608 | 20.00 | 59.62 | 34.80 | 60.67 | 42.71 | 55.06 |
| base=10000 | 26.68 | 61.57 | 35.09 | 62.79 | 42.03 | 55.78 |
| base=40000 | 26.87 | 60.80 | 35.53 | 60.49 | 42.37 | 55.26 |
| base=80000 | 26.76 | 61.25 | 35.38 | 61.20 | 41.36 | 55.32 |
| base=160000 | 27.06 | 61.22 | 35.96 | 60.14 | 42.03 | 54.35 |
| base=400000 | 27.53 | 60.94 | 35.82 | 62.96 | 42.37 | 54.01 |
| base=600000 | 26.81 | 61.28 | 35.67 | 61.20 | 42.03 | 53.75 |
| base=1000000 | 26.51 | 59.22 | 35.53 | 60.85 | 42.37 | 52.94 |
| base=500, 16K | 11.86 | 44.74 | 35.38 | 49.21 | 33.22 | 38.34 |
| base=10000, 16K | 19.14 | 51.22 | 34.65 | 58.20 | 41.02 | 51.11 |
| base=40000, 16K | 23.68 | 57.32 | 35.53 | 59.61 | 41.69 | 53.68 |
| base=80000, 16K | 24.90 | 58.74 | 34.94 | 61.38 | 38.98 | 52.86 |
| base=120000, 16K | 25.37 | 58.16 | 34.65 | 61.90 | 41.02 | 52.93 |
| base=1000000, 16K | 24.35 | 57.26 | 35.53 | 59.44 | 39.66 | 51.61 |

Table 4: Continued table of Table3. The final score of MMLU is the average score of all subtasks.

## B.3 LONG CONTEXT VALIDATION

Besides, to validate that our proposed framework can improve the long-range real-world tasks, we also compare the close-ended subtasks included in L-Eval benchmark(An et al., 2023) within 16K context length, including the 16-shot GSM8K(Cobbe et al., 2021), TOEFL-QA(Tseng et al., 2016), TopicRetrieval(Li et al., 2023a), QuALITY(Pang et al., 2022), and Coursera(An et al., 2023). Our method is still capable of these tasks, as shown in Table 5. For RoPE-based LLMs fine-tuned on pre-training context, namely 4K tokens, they still have certain effects on these tasks with 16K context

size, which proves one of the claims of our work, that tuning with the original pre-training context can also boost the length extrapolation. This proves that we extrapolate the RoPE-based LLM to a longer context without the disturbance of domain distribution in pre-trained data concerning the change of training context length (Peng et al., 2023). We also report the evaluation results of RoPE-based LLM fine-tuned with longer context, such as 16K tokens, in Table 3, Table 4 and Table 5 and the length extrapolation improvement will be further analyzed in Appendix C.

|  | GSM | TOEFL | TopicRet | QuALITY | Coursera |
|---|---|---|---|---|---|
| LLaMA2 7B | 0.00 | 34.94 | 0.00 | 3.96 | 0.00 |
| NTK fixed $\alpha = 8$ | 5.00 | 34.20 | 31.33 | 34.16 | 21.51 |
| NTK dynamic | 9.00 | 37.17 | 24.67 | 30.20 | 22.09 |
| base=500 | 4.00 | 25.65 | 0.00 | 18.32 | 19.19 |
| base=500, log-scaled | 3.00 | 25.65 | 0.00 | 18.32 | 19.77 |
| base=652 | 6.00 | 16.73 | 0.00 | 13.37 | 11.63 |
| base=652, log-scaled | 6.00 | 16.73 | 0.00 | 16.83 | 11.63 |
| base=10000 | 0.00 | 18.59 | 0.00 | 2.97 | 0.00 |
| base=40000 | 13.00 | 34.20 | 14.67 | 22.77 | 11.63 |
| base=80000 | 11.00 | 36.43 | 29.33 | 23.76 | 15.70 |
| base=160000 | 10.00 | 40.15 | 38.00 | 22.28 | 16.28 |
| base=400000 | 8.00 | 41.64 | 36.67 | 22.77 | 15.70 |
| base=600000 | 12.00 | 38.66 | 34.00 | 25.25 | 16.86 |
| base=1000000 | 9.00 | 31.97 | 34.00 | 21.29 | 16.28 |
| base=2000000 | 7.00 | 30.86 | 31.33 | 22.77 | 15.70 |
| base=500, 16K | 2.00 | 25.65 | 0.00 | 16.83 | 20.35 |
| base=10000, 16K | 5.00 | 42.38 | 0.00 | 19.80 | 13.95 |
| base=40000, 16K | 11.00 | 43.49 | 20.67 | 28.22 | 19.19 |
| base=80000, 16K | 12.00 | 43.49 | 35.33 | 30.20 | 21.51 |
| base=120000, 16K | 14.00 | 43.12 | 40.00 | 26.24 | 20.93 |
| base=1000000, 16K | 13.00 | 44.24 | 41.33 | 25.74 | 21.51 |
| LLaMA2 13B | 0.00 | 59.48 | 0.00 | 6.44 | 0.00 |
| NTK fixed $\alpha = 8$ | 8.00 | 58.74 | 36.00 | 35.64 | 23.26 |
| NTK dynamic | 22.00 | 59.85 | 30.00 | 35.15 | 28.49 |
| base=500 | 9.00 | 37.17 | 0.00 | 30.20 | 25.58 |
| base=500, log-scaled | 11.00 | 37.17 | 0.00 | 28.71 | 25.58 |
| base=652 | 12.00 | 37.17 | 0.00 | 30.69 | 25.58 |
| base=652, log-scaled | 9.00 | 37.17 | 0.00 | 31.19 | 25.00 |
| base=10000 | 0.00 | 56.13 | 0.00 | 7.43 | 0.00 |
| base=40000 | 18.00 | 64.68 | 20.67 | 49.01 | 16.86 |
| base=80000 | 20.00 | 63.94 | 30.00 | 52.97 | 24.42 |
| base=160000 | 23.00 | 62.83 | 39.33 | 51.49 | 24.42 |
| base=400000 | 17.00 | 59.48 | 34.67 | 51.49 | 29.07 |
| base=600000 | 14.00 | 63.94 | 36.00 | 47.52 | 29.65 |
| base=1000000 | 10.00 | 64.68 | 28.67 | 45.05 | 19.19 |
| base=500, 16K | 6.00 | 20.07 | 0.00 | 21.29 | 26.74 |
| base=10000, 16K | 13.00 | 55.39 | 0.00 | 38.61 | 25.00 |
| base=40000, 16K | 25.00 | 63.94 | 19.33 | 49.50 | 18.60 |
| base=80000, 16K | 21.00 | 63.20 | 34.67 | 40.10 | 15.12 |
| base=120000, 16K | 22.00 | 63.94 | 40.67 | 39.11 | 19.19 |
| base=1000000, 16K | 9.00 | 63.57 | 33.33 | 48.02 | 16.86 |

Table 5: Long context validation results of scaling RoPE with different base and other extrapolating methods in LLaMA2 (Touvron et al., 2023b). The first box is the result of 7B size and the second box is those of 13B version.

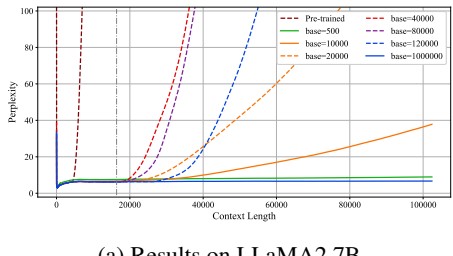

(a) Results on LLaMA2 7B

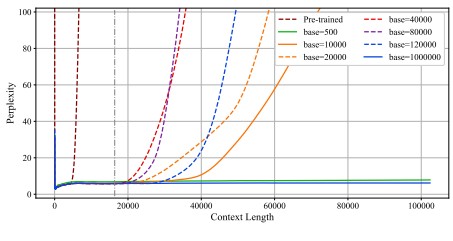

(b) Results on LLaMA2 13B

Figure 9: Perplexity of RoPE fine-tuned with 16K context length and smaller or larger bases on the validation data of Books3 (Presser, 2020). Surprisingly, as the base increases from 500 to 1000000, the extrapolation capability first becomes weaker and then becomes stronger.

## C  EXTENSION

Concerning that the current extrapolation methods in the fine-tuning phase dominantly apply longer tuning contexts, we propose an extended version of the scaling law of RoPE-based extrapolation.

**Theorem 3.** *(Extended Scaling Law of RoPE-based Extrapolation)*     For RoPE-based LLMs pre-trained with context length $T_{\text{train}}$ and critical dimension $d_{\text{extra}}$, if we adjust the base to $\beta$ and then conduct fine-tuning with context length $T_{\text{tune}} \geq T_{\text{train}}$, the extrapolation performance of RoPE-based LLMs will get improved. Importantly, there exists a *critical base* $\beta_0$ decided by $T_{\text{tune}}$ and $T_{\text{train}}$.

$$\beta_0 = 10000^{\log_{\frac{T_{\text{train}}}{2\pi}} \frac{T_{\text{tune}}}{2\pi}}. \tag{12}$$

If $\beta > \beta_0$, the extrapolation upper bound is decided by $\beta$ and $d_{\text{extra}}$ as follows:

$$T_{\text{extra}} = 2\pi \cdot \beta^{d_{\text{extra}} \cdot \frac{1}{d}} = 2\pi \cdot \beta^{\left\lceil \frac{d}{2} \log_{10000} \frac{T_{\text{train}}}{2\pi} \right\rceil \cdot \frac{2}{d}}. \tag{13}$$

Otherwise, the extrapolation upper bound is $T_{\text{tune}}$ and the critical dimension is updated satisfying

$$d'_{\text{extra}} = 2 \left\lceil \frac{d}{2} \log_\beta \frac{T_{\text{tune}}}{2\pi} \right\rceil \geq 2 \left\lceil \frac{d}{2} \log_{10000} \frac{T_{\text{train}}}{2\pi} \right\rceil = d_{\text{extra}}. \tag{14}$$

However, the extrapolation beyond $T_{\text{tune}}$ will acquire further enhancement if $\beta$ gets smaller. Particularly, when $\beta$ is smaller than $\beta_1, \beta_2, \beta_3$ as follows, the enhancement will be more significant.

$$\beta_1 = \frac{2T_{\text{tune}}}{\pi}, \quad \beta_2 = \frac{T_{\text{tune}}}{\pi}, \quad \beta_3 = \frac{T_{\text{tune}}}{2\pi}. \tag{15}$$

Theorem 3. serves as both a combination and generalization of Theorem 1., Lemma 1. and Theorem 2. Here, the critical base is the worst base for extrapolation as well as the smallest base forcing RoPE to extrapolate depending on the feature dimensions within the critical dimension. Specifically, when $T_{\text{tune}} = T_{\text{train}}$, the critical base, $\beta_0 = 10000$, relates to the intriguing observation outlined in Section 2. Equation 13 corresponds to tuning RoPE with larger bases discussed in Section 3.3, Equation 14 corresponds to the definition of critical dimension in Section 3.2 and Equation 15 corresponds to tuning RoPE with smaller bases discussed in Section 3.1. If $T_{\text{tune}} > T_{\text{train}}$, RoPE-based LLMs can accommodate a broader context window. As illustrated in Figure 9, when fine-tuning LLaMA2 7B and 13B (Touvron et al., 2023b) within a 16K context, the max context length exceeds or equals 16K, surpassing the original LLaMA2 7B and 13B respectively, regardless of the base value.

From a periodic perspective, since both base and training length have changed, it prompts us to consider whether additional positional information has been integrated during the fine-tuning phase for feature dimensions beyond the critical dimension, namely the 92nd dimension for LLaMA2 (Touvron et al., 2023b), as indicated in Equation 5. According to the definition of the critical dimension, based on the current base $\beta$, we can calculate how many dimensions the RoPE-based LLM has where $\sin$ and $\cos$ complete a period within the tuning length. If base $\beta > \beta_0$, then $\tilde{d}_{\text{extra}}$, the

number of dimensions that cover a period during fine-tuning have already been able to traverse a complete period during pre-training, given that

$$
\begin{aligned}
\tilde{d}_{\text{extra}} = 2 \left\lceil \frac{d}{2} \log_{\beta} \frac{T_{\text{tune}}}{2\pi} \right\rceil &\leq 2 \left\lceil \frac{d}{2} \log_{\beta_0} \frac{T_{\text{tune}}}{2\pi} \right\rceil = 2 \left\lceil \frac{d}{2} \log_{10000^{\log_{\frac{T_{\text{train}}}{2\pi}} \frac{T_{\text{tune}}}{2\pi}}} \frac{T_{\text{tune}}}{2\pi} \right\rceil \\
&= 2 \left\lceil \frac{d}{2} \frac{1}{\log_{\frac{T_{\text{train}}}{2\pi}} \frac{T_{\text{tune}}}{2\pi}} \log_{10000} \frac{T_{\text{tune}}}{2\pi} \right\rceil = 2 \left\lceil \frac{d}{2} \frac{\log_{\frac{T_{\text{tune}}}{2\pi}} \frac{T_{\text{train}}}{2\pi}}{\log_{\frac{T_{\text{tune}}}{2\pi}} 10000} \right\rceil . \\
&= 2 \left\lceil \frac{d}{2} \log_{10000} \frac{T_{\text{train}}}{2\pi} \right\rceil = d_{\text{extra}}
\end{aligned}
$$

Therefore, the critical dimension remains unchanged. Referring to Theorem 2 in Section 3.3, we can calculate the extrapolation upper bound based on the updated base and the original critical dimension as Equation 13, exactly the same as Equation 5. For LLaMA2 (Touvron et al., 2023b) fine-tuned with a 16K context, as illustrated in Figure 9, the critical base is 71738, given Equation 12. For bases greater than 71738, such as 80000, 120000, and 1000000, their extrapolation upper bounds surpass 16K and the larger base corresponds to a longer context, corroborating our theoretical framework.

If base $\beta \leq \beta_0$, then during the fine-tuning phase, the number of dimensions able to complete a period surpasses the original critical dimension, so the critical dimension is updated as Equation 14. Besides, since this dimension depends on the fine-tuning length, the extrapolation upper bound is still constrained within the fine-tuning length $T_{\text{tune}}$. However, if $\beta$ is so small that the input of every $\cos(t-s)\theta_n, \sin(t-s)\theta_n$ can span values from 0 to $\pi/2$, $\pi$, or $2\pi$ within the fine-tuning length $T_{\text{tune}}$, as indicated by Equation 15, similar to Theorem 1 in Section 3.1, then the extrapolation performance will get further improved, marked by a more stable perplexity growth curve. For LLaMA2 (Touvron et al., 2023b) fine-tuned with a 16K context, as shown in Figure 9, for bases smaller than the critical base 71738, such as 60000, 20000, 10000, and 500, the performance curves become progressively more stable. Among them, although $\beta = 10000$ performs poorly in fine-tuning at the original context length, the performance gets significantly improved this time because the inputs of $\cos$ or $\sin$ have traversed to $\pi/2$ within the 16K context length. When $\beta = 500$, LLaMA2 achieved a similar terrific performance as $\beta = 1000000$, namely the design of Code LLaMA (Rozière et al., 2023), a context length with at least 100K tokens, breaking the curse of entropy explosion mentioned in Han et al. (2023). Since there exists an upper bound for extrapolation based on $\beta = 1000000$, RoPE tuned with base 500 on 16K context length has the potential to extrapolate to an infinite context, thus answering Q3 in the Introduction.

Similarly, we also use scatter plots to visualize the fluctuation of attention scores for different bases after fine-tuning with a 16K context as illustrated in Figure 10. For base 500, given its exposure to enough fluctuations during the training phase, the perplexity curve remains notably stable. For base 10000, it is clear that the fluctuation of attention scores in the last 36 dimensions is somewhat limited, leading to a noticeable improvement in extrapolation performance given Equation 15. For base 40000, the position information acquired in the fine-tuning phase shrinks further as the base increases. For base 120000, the critical dimension goes back to 92 dimensions, and the extrapolation performance is governed by the first 92 dimensions. Remarkably, the extrapolation upper bound given Equation 13 matches the maximum supported context length. For base 1000000, the period of the first 92 dimensions is further extended, corresponding to a context length expanding beyond 100K. Eventually, based on the above interpretation, we validate the correctness of Theorem 3. and provide a unified framework to explain the extrapolation of RoPE-based LLM with arbitrary base and fine-tuning context length.

# D DISCUSSION

Besides, we discuss the instructive value of our theory for other extrapolation strategies focused on achieving longer context during the testing phase. These methods are still necessary given two facts. On one hand, the performance of RoPE with a smaller base is still left behind compared with RoPE with much larger bases, such as 1000000, as shown in Figure 1. On the other hand, for RoPE with a base that is not large enough, it still can not extrapolate to a context of 100K or longer as shown in Figure 2. In order to further enhance RoPE's adaptability to a longer context, whatever the base

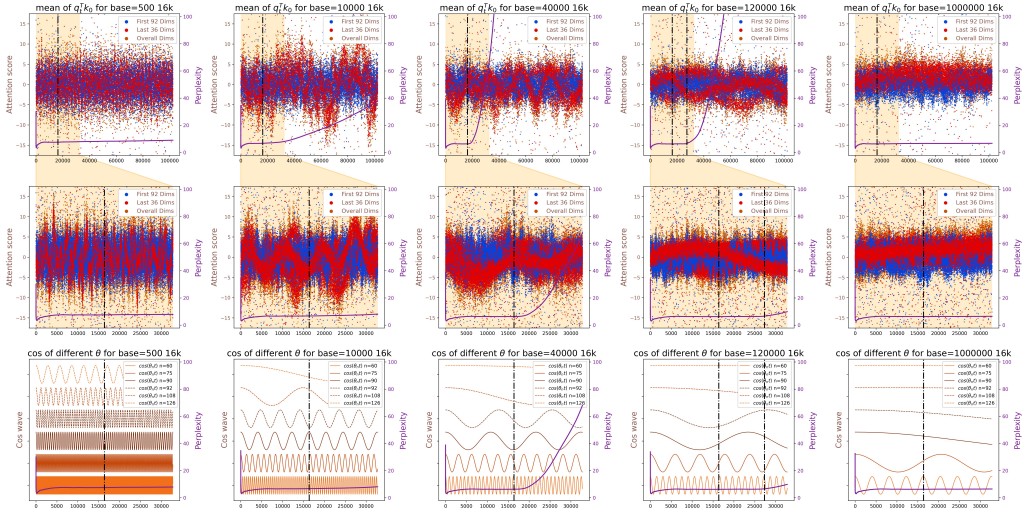

Figure 10: The relation between attention scores in first 92 and last 36 dimensions with the extrapolation performance in LLaMA 7B (Touvron et al., 2023b) evaluated or fine-tuned with different bases at 16K context length. The meaning of each row is the same as that in Figure 6 except that the second row highlights the changes in the first 32K tokens instead of the first 16K tokens.

value is, we discuss the effect of two complementary methods in the inference phase, log-scaled attention (Su, 2023b) and dynamic-scaled RoPE (bloc97, 2023a) on RoPE with different bases.

**Log-scaled Attention** as shown in Equation 16, is a classic technique originally raised in Chiang & Cholak (2022) and currently applied in RoPE-based extrapolation (Su, 2023b; Alibaba, 2023). It involves multiplying the original attention matrix by the logarithm of the current inference length $t$. Traditionally, the base of the logarithm is training length $T_{\text{train}}$. However, given Equation 13 in Theorem 3., the attention score within the max supported context length $T_{\text{extra}}$ is reliable. So we take $T_{\text{extra}}$ as the logarithm base and set the lower limit for the logarithmic correction value as 1, meaning that no additional log scale is required within the extrapolation upper bound.

$$\boldsymbol{A}_{t,s} = \text{Re}\left[\sum_{n=0}^{d/2-1} \tilde{q}_t^{(n)} \tilde{k}_s^{(n)*} e^{i(t-s)\theta_n}\right] \cdot p_t.$$

$$p_t = \max\left(1, \log_{T_{\text{extra}}} t\right)$$

(16)

Besides Log-scaled attention, window method, such as sliding window and its variant, is also a widely accepted strategy for extrapolation, used in inference or evaluation (Press et al., 2022; Sun et al., 2023). Compared with the above strict window-based method, we follow the xPos method proposed in Sun et al. (2023), shown in Equation 17, originally used in the pre-training phase. In this work, we regard this method as a soft sliding window used in the inference phase as use it as a further complement to the log-scaled method. Still, we do little modification besides using the $T_{\text{extra}}$ as the denominator instead of the original denominator $T_{\text{train}}$.

$$\boldsymbol{A}_{t,s} = \text{Re}\left[\sum_{n=0}^{d/2-1} \tilde{q}_t^{(n)} \tilde{k}_s^{(n)*} \zeta_n^{\frac{t-s}{T_{\text{extra}}}} e^{i(t-s)\theta_n}\right].$$

$$\zeta_n = \frac{\gamma + 2n/d}{\gamma + 1}, \; n = 0 \cdots \frac{d}{2} - 1, \; \gamma = 0.4$$

(17)

**Dynamic-scaled RoPE** namely Dynamic NTK (bloc97, 2023a) is a widely used extrapolation. Here, we only do two little modifications. One is to change the base 10000 in vanilla RoPE (Su et al., 2021) with the base scaled in the fine-tuning phase, $\beta$. The other is still to replace the $T_{\text{train}}$ in Equation **??** with $T_{\text{extra}}$ we derive given Equation 13 in Theorem 3.

$$\text{for } \boldsymbol{A}_{t,s}, \theta_n = (\beta \cdot \alpha_t)^{-2n/d}, \text{ where } \alpha_t = \max\left(1, 2^{\left\lceil \log_2 \frac{t}{T_{\text{extra}}} \right\rceil + 1} - 1\right).$$

(18)

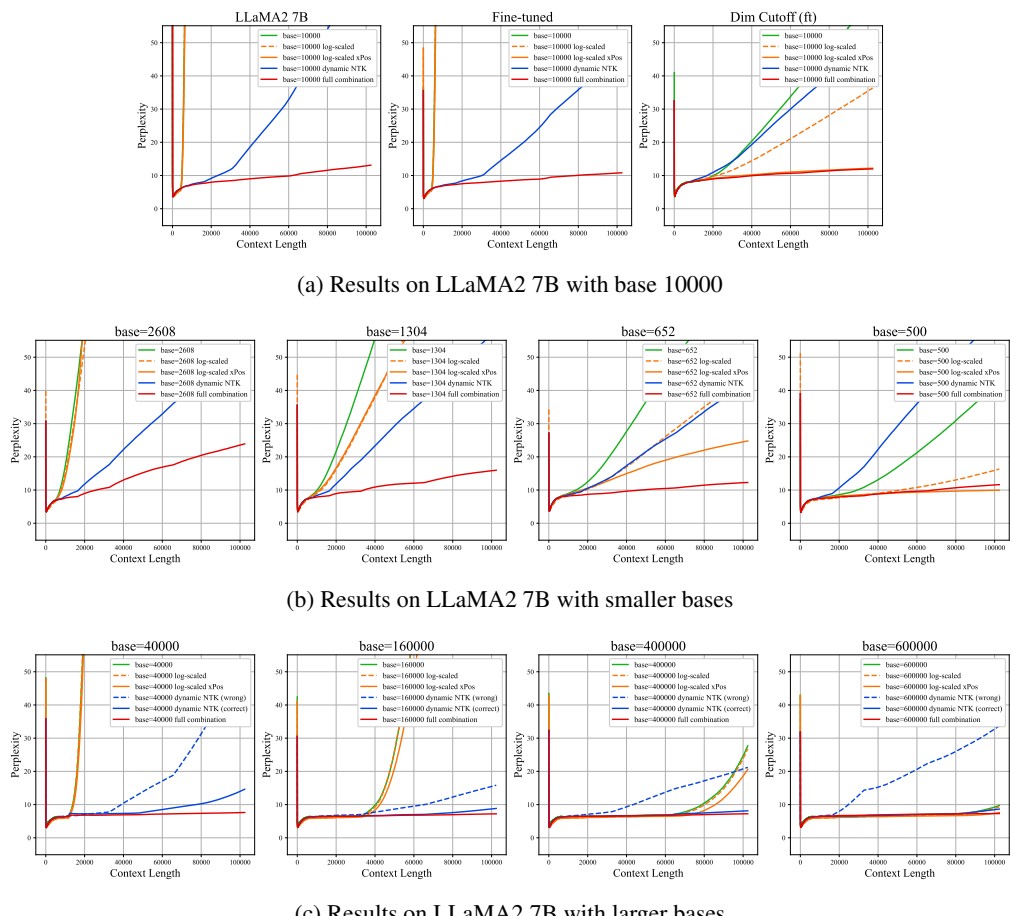

(a) Results on LLaMA2 7B with base 10000

(b) Results on LLaMA2 7B with smaller bases

(c) Results on LLaMA2 7B with larger bases

Figure 11: Perplexity on the validation data from Books3 dataset (Presser, 2020) of LLaMA 7B (Touvron et al., 2023b) based on RoPE with different bases enhanced with log-scaled attention, xPos and dynamic-scaled RoPE and their combination. Here, *wrong* means calculating $\alpha_t$ in Dynamic NTK (bloc97, 2023a) or $p_t$ in (Su, 2023b) with training length $T_{\text{train}}$, while *correct* means using extrapolation upper bound $T_{\text{extra}}$ instead of $T_{\text{train}}$.

We experiment with these two methods on LLaMA2 7B (Touvron et al., 2023b) and get the results as shown in Figure 11. Figure 11a shows the results on LLaMA2 based on RoPE with base 10000. It is clear that both log-scaled attention almost does nothing for pre-trained and fine-tuned LLaMA2, but show great improvement for fine-tuning LLaMA2 with cutting the last 36 dimensions in $\boldsymbol{q}_t, \boldsymbol{k}_s$ off. This phenomenon further proves that the fluctuation coming from the dimensions of $\boldsymbol{q}_t, \boldsymbol{k}_s$ beyond the critical dimension is the root cause of the extrapolation problem of RoPE-based LLM.

Figure 11b shows the results on LLaMA2 based on RoPE with bases smaller than 10000. There is a clear trend that with the reduction of the base value, the improvement obtained from the log-scaled attention is more dominant while the effect of Dynamic NTK shrinks gradually. For RoPE with base 500, the perplexity curve of log-scaled RoPE is flat enough, indicating the extrapolation capability to support 100K context length. On the contrary, Dynamic NTK shows a clear side effect. Hence, the position information learned in the training phase is reliable enough for LLM to extrapolate further, corresponding to Theorem 1. in Section 3.1.

Figure 11c shows the results on LLaMA2 based on RoPE with bases larger than 10000, such as 40000, 160000, 400000, and 600000. We do not test the performance of two methods on RoPE with base 1000000, since it already achieved the context length of 100K. Here, we enable RoPE with bases larger than 10000 and smaller than 1000000 to extrapolate beyond the context length of 100K in the inference phase. For RoPE with larger bases, the improvement of extrapolation performance

obtained from Dynamic NTK is more remarkable. The working principle of Dynamic NTK has been visualized in Figure 8 and discussed in Section 3.4.

Besides, replacing the $T_{\text{train}}$ with $d_{\text{extra}}$ becomes significantly important for RoPE with larger bases. For example, if Dynamic NTK is carried out based on $T_{\text{train}}$, the improvement will be limited and even destroyed when the base is large enough like 400000 and 600000. This phenomenon proves the guidance value of this work for other extrapolation works. In conclusion, for a base smaller than $\beta_3$ defined in Equation 1, every dimension has learned complete positional information. Then the log-scaled method is sufficient to enhance extrapolation. For a base larger than $\beta_0$, namely 10000 for tuning on the original context, Dynamic NTK in the correct way is a good helper for extrapolation to a much longer context.

## E  TEXT CONTINUATION

Finally, we execute a text continuation as our case study. For LLaMA2 7B(Touvron et al., 2023b) fine-tuned with various base values and fine-tuning lengths, we provide a context of 32K tokens from Books3(Presser, 2020) and prompt the model to continue writing. The resulting texts, after cleaning the special characters, are shown in Figure 12. Remarkably, whether the base is set at 500 or 1000000, the generated text remains coherent, grammatically accurate, and logically consistent. For base 500, extended tuning length or incorporating log-scaled attention Su (2023b) in the testing phase yields improved continuation results. Given the strict extrapolation upper bound for base 1000000, there is a compelling case that a model fine-tuned with base 500 possesses an infinite extrapolation potential.

## LIMITATION

In this work, our primary objective is to elucidate the mechanisms by which RoPE, using either smaller or larger bases, enhances extrapolation performance. Additionally, there remains a need to find whether there exist upper and lower bounds for adjusting the base. Besides, it needs additional efforts to integrate the performance of RoPE with smaller bases in the pre-training stage. Finally, since the base value and pre-training context length are keys to extrapolation, there might be an optimal pair of base and training length for RoPE-based LLM pre-training to be discovered.

**Context:** THIS BOOK IS FOR INFORMATIONAL PURPOSES ONLY. IT IS NOT IN-TENDED TO BE A SUBSTITUTE FOR CONSULTATION WITH A PROFESSIONAL HEALTHCARE PRACTITIONER. ...... They invited airplane pilots with an average age of fifty-two who were interested in participating and evaluated performance in flight simulations. Every three minutes, the pilots were peppered with complex commands and codes that had to be remembered and implemented while under stressful flying conditions. After seven simulations, the study subjects were given Aricept for

**Reference:** a month and then retested under equally demanding but new simulations. The results showed that they performed better in the later round of evaluations, and the authors stated that Aricept "appears to have beneficial effects on retention of training on complex aviation tasks in nondemented older adults," and these findings were ultimately published in Neurology. There is already a major pharmacologic market for physical performance-enhancing drugs. With an aging population, interest in medications for cognitive enhancement will likely grow.

**Output (LLaMA2 7B):** OOOOOOOOOOOOOOOOOOOOOOOOOOOOOOOOOOOOOO

**Output (base=500):** the purpose of improving their cognitive performance. Neuro-Geek: Alzheimer's Drugs for the Healthy? While most adults will not develop Alzheimer's, many adults will have mild cognitive impairments that affect how they perform in daily life and work. In the sector of the scientific community focused on developing drugs to improve the symptoms associated with dementia, there is an understanding that those same drugs may be used by healthy people with the goal of maintaining cognitive acuity or even, in certain circumstances, improving mental agility above their established baseline.

**Output (base=1000000):** a month. The results were clear: the drug improved their performance. The study was published in 2015, and the authors concluded that "the results suggest that the drug may be able to improve cognitive performance in healthy individuals." The study was small, and the results are not yet conclusive. But it is a start. Prescription Antidepressants The first antidepressants were developed in the 1950s, and they have been used to treat depression ever since. But in the 1990s, scientists began to wonder if they might also be useful for improving cognitive functioning.

**Output (base=500 16K):** a week. The drug is a cholinesterase inhibitor, which means it blocks the breakdown of acetylcholine, a neurotransmitter that helps regulate memory and learning. The results were striking. The pilots who took the drug performed better than those who didn't. They were able to remember more of the complex commands and codes, and they were able to implement them more quickly. The study authors concluded that the drug "may be useful in improving cognitive performance in healthy individuals." But there's a catch. The drug is not approved for use in healthy people. It's only approved for people with Alzheimer's disease.

**Output (base=1000000 16K):** a month, then switched to a placebo for another month. The results were striking. The pilots who took the drug improved their performance by 20 percent, while those who took the placebo got worse. The study was small, and the results were not replicated in a larger study. But the idea that healthy people might be able to improve their cognitive performance with drugs designed to treat dementia is intriguing. Prescription Antidepressants The most widely prescribed class of antidepressants, selective serotonin reuptake inhibitors (SSRIs), are also used to treat anxiety, obsessive-compulsive disorder, and panic disorder.

**Output (base=500 16K log-scaled):** a week. The drug is a cholinesterase inhibitor, which means it blocks the breakdown of acetylcholine, a neurotransmitter that is critical for memory and learning. The results were striking. The pilots who took the drug performed better than those who didn't. They were able to remember more of the complex commands and codes and to implement them more quickly. But the study also had a major flaw. The pilots were not randomly assigned to take the drug or not. They were all given the drug because they were already healthy. The study's authors concluded that the drug improved performance in pilots who were already healthy. But that's not what the study actually showed.

Figure 12: Text continuation after a 32K-length context from Books3 (Presser, 2020) of LLaMA2 7B(Touvron et al., 2023b) tuned with base 500 or 1000000 and context length 4K or 16K. It shows that our methods could still produce cohesive and fluid output text in a much longer context.

