# OpenReview forum: "Scaling Laws of RoPE-based Extrapolation"
_ICLR.cc/2024/Conference — ICLR 2024 poster_

### Official Review · Reviewer_Dcc1 · 2023-10-18

**Soundness:** 2 fair
**Presentation:** 3 good
**Contribution:** 3 good
**Rating:** 8
**Confidence:** 5

**Summary:**

This paper studies how shall we scale the transformer context length up via modifying RoPE. Different from popular approaches that increase the rotary base, authors found both increasing and decreasing the rotary base could significantly enhance its extrapolation performance.

**Strengths:**

1) This paper studies a very interesting problem: How shall we modify the 10^4 rotary base?
2) Well-written.
3) Some reasoning and insights are very cool. Such as the Q1 and Q2 in the introduction.
4) The theoretical analysis of the benefits of a smaller rotary base would be helpful to the long seq community.

**Weaknesses:**

1) I have one strong concern about the evaluation of this paper. This paper only evaluates the scaled model on the long-context perplexity and short-context well-established benchmarks. Low ppl on long context language modeling cannot really reflect the good results on long-range reasoning. The short-context results can only show the proposed approach is not that harmful on short tasks. If so, there is no evidence showing that the proposed framework can really improve the long-range real-world tasks, which is exactly what we espect in the long-range modeling research.

2) There is no related work section. It is "okay" for expert readers, but it would be very helpful to make a broader impact if authors can write one. I think authors should at least include:
- scaling seq len with position embedding. The authors discussed some of these approaches, but it would be better to summarize them in a thread.
- scaling seq len with efficient implementation, such as Flash attention 1&2 [1]
- scaling seq len with a distributed system, such as Sequence Parallelism [2][3]


[1] https://github.com/Dao-AILab/flash-attention

[2] Li, Shenggui, et al. "Sequence parallelism: Long sequence training from system perspective." arXiv preprint arXiv:2105.13120 (2021).

[3] Jacobs, Sam Ade, et al. "DeepSpeed Ulysses: System Optimizations for Enabling Training of Extreme Long Sequence Transformer Models." arXiv preprint arXiv:2309.14509 (2023).

3) It would also be very interesting to study the filling-in-the-middle pattern after applying the proposed trick. Is the filling-in-the-middle related to the position embedding? This is just a bonus question. It is an okay paper without this study, but I suggest authors to study this to deliver more insights.

I give a relatively negative score at this stage because of the lack of real-world task evaluation, but the insights of this paper are interesting to me. Authors need to further justify them. I can improve my score if the authors can add the real-world long-context evaluation into the draft.

**Questions:**

See weakness above.

---

> ### Author Response · Authors · 2023-11-15
> **Response (1/3)**
>
> Dear reviewer, thank you for your review. When it comes to long-range real-world tasks, we compared some tasks often used in long-context research [1-2]. On one hand, concating multi-shot short prompts into a long context is included, such as 25-shot ARC-challenge [3], 10-shot Hellaswag [4], and 16-shot GSM8K [5] in LEval [2]. On the other hand, QA and extraction tasks of long documents are also involved, such as the TOEFL-QA [6] and TopicRetrieval [7] in L-Eval [2]. Our method is still capable of these tasks, as shown in the table below. Particularly, even for RoPE-based LLMs fine-tuned on pre-training context, namely 4k tokens, they still have certain effects on these tasks with 16K context size. This also proves one of the claims of our work, that tuning with the original pre-training context can also boost the length extrapolation of RoPE-based LLMs. We will publish more evaluation results for long document tasks in the later version.
>
> |  |  | 25-shot ARC-c | 10-shot hellaswag | 10-shot GSM | TOEFL-QA | topic_retrieval |
> | --- | --- | --- | --- | --- | --- | --- |
> | llama2_7B | pretrained | 48.14  | 72.78  | 0.00  | 34.94  | 0.00  |
> |  | ntk_fixed_8 | 46.78  | 68.94  | 5.00  | 34.20  | 31.33  |
> |  | ntk_dynamic | 48.14  | 72.73  | 9.00  | 37.17  | 24.67  |
> |  | base_10000 | 47.80  | 72.49  | 0.00  | 18.59  | 0.00  |
> |  | base_80000 | 47.12  | 72.66  | 11.00  | 36.43  | 29.33  |
> |  | base_160000 | 45.42  | 72.60  | 10.00  | 40.15  | 38.00  |
> |  | base_400000 | 47.12  | 72.63  | 8.00  | 41.64  | 36.67  |
> |  | base_600000 | 46.10  | 72.54  | 12.00  | 38.66  | 34.00  |
> |  | base_1000000 | 46.78  | 72.71  | 9.00  | 31.97  | 34.00  |
> |  | base_2000000 | 46.10  | 72.57  | 7.00  | 30.86  | 31.33  |
> | llama2_13B | pretrained | 54.58  | 73.89  | 0.00  | 59.48  | 0.00  |
> |  | ntk_fixed_8 | 50.85  | 66.88  | 8.00  | 58.74  | 36.00  |
> |  | ntk_dynamic | 54.58  | 73.93  | 22.00  | 59.85  | 30.00  |
> |  | base_10000 | 53.56  | 75.53  | 0.00  | 56.13  | 0.00  |
> |  | base_80000 | 53.56  | 75.69  | 20.00  | 63.94  | 30.00  |
> |  | base_160000 | 53.56  | 75.50  | 23.00  | 62.83  | 39.33  |
> |  | base_400000 | 54.24  | 75.56  | 17.00  | 59.48  | 34.67  |
> |  | base_600000 | 53.90  | 75.29  | 14.00  | 63.94  | 36.00  |
> |  | base_1000000 | 52.54  | 75.18  | 10.00  | 64.68  | 28.67  |

---

> ### Author Response · Authors · 2023-11-15
> **Response (2/3)**
>
> When it comes to the related work, if the paper gets accepted, we will summarize scaling seq len with position embedding into one subsection and add a subsection respectively to scaling seq len with efficient attention and scaling seq len with a distributed system. The added content is as follows.
>
> Long sequence modeling faces the following three challenges: the O(N2) computational complexity, the difficulty of parallelism for sequence length, and the collapse of the performance beyond the training length. First of all, due to the O(N2) complexity of the self-attention mechanism in Transformers, long sequence modeling requires a large computational overhead. Early efficient Transformers reduced computational overhead through methods such as recalculation like ReFormer [8] and sparse attention such as LongFormer [9] and BigBird [10]. After that, LinFormer [11] linearizes the self-attention complexity through low-rank approximation, and Performer [12] expands the softmax calculation through a kernel function. Compared with these efficient attention works based on approximation, FlashAttention [13-14] reduces HBM reading and writing and turns self-attention into a form that can be iteratively calculated in a loop, which greatly improves the utilization of computing resources, reduces the cost of long sequence calculations. and has become the standard for long-sequence modeling.
>
> The second challenge is parallelism for sequence length, that is, Sequence Parallelism (SP) [15]. Since the memory and calculation of a single device are limited, ColossalAI-SP [15] first proposes the segmentation and parallelism for the sequence dimension in addition to tensor parallelism for the hidden dimension and pipeline parallelism for model depth. On this basis, Ring Attention [16] uses blockwise self-attention to split long sequences into different devices and overlap the communication of key-value blocks. Besides, LightSeq [17] further improves the efficiency of long sequence modeling through load balancing for causal language modelings and a re-materialization-aware checkpointing strategy. The advantage of the above sequence parallelism is that it can break through architectural limitations and achieve infinitely long sequence modeling. In contrast, another type of sequence parallelism emphasizes maintaining the combination with existing efficient self-attention mechanisms such as FlashAttention [13-14] to achieve input with almost infinity. For example, Megatron-SP [18] only uses sequence parallelism during Dropout and Layernorm operations, thereby reducing more activation redundancy. In addition, DeepSpeed-Ulysses [19] uses an alternative all-to-all collective communication gathering weight for attention computation when segmenting the sequence, avoiding communication overhead that originally increases with length.

---

> ### Author Response · Authors · 2023-11-15
> **Response (3/3)**
>
> When it comes to the filling-in-the-middle pattern [20], filling-in-the-middle [20] is a training paradigm that we believe is orthogonal to RoPE-based extrapolation. If you talk about lost-in-the-middle [21], the relationship between lost-in-the-middle and RoPE-based extrapolation is an interesting research topic. Preliminarily, we find that tuning with a larger base is essentially enlarging the interval lost span in the middle. In essence, the RoPE-based LLMs still only pay attention to the information of the adjacent and initial positions, as discussed in StreamingLLM [22]. We are now conducting more experimental verifications on this.
>
> [1] Peng, Bowen, et al. "Yarn: Efficient context window extension of large language models." arXiv preprint arXiv:2309.00071 (2023).
>
> [2] An, Chenxin, et al. "L-Eval: Instituting Standardized Evaluation for Long Context Language Models." arXiv preprint arXiv:2307.11088 (2023).
>
> [3] Clark, Peter, et al. "Think you have solved question answering? try arc, the ai2 reasoning challenge." arXiv preprint arXiv:1803.05457 (2018).
>
> [4] Zellers, Rowan, et al. "Hellaswag: Can a machine really finish your sentence?." arXiv preprint arXiv:1905.07830 (2019).
>
> [5] Cobbe, Karl, et al. "Training verifiers to solve math word problems." arXiv preprint arXiv:2110.14168 (2021).
>
> [6] Tseng, Bo-Hsiang, et al. "Towards machine comprehension of spoken content: Initial TOEFL listening comprehension test by machine." arXiv preprint arXiv:1608.06378 (2016).
>
> [7] Li, Dacheng, et al. How long can open-source llms truly promise on context length?, 2023. URL https://lmsys.org/blog/2023-06-29-longchat.
>
> [8] Kitaev, Nikita, Łukasz Kaiser, and Anselm Levskaya. "Reformer: The efficient transformer." arXiv preprint arXiv:2001.04451 (2020).
>
> [9] Beltagy, Iz, Matthew E. Peters, and Arman Cohan. "Longformer: The long-document transformer." arXiv preprint arXiv:2004.05150 (2020).
>
> [10] Zaheer, Manzil, et al. "Big bird: Transformers for longer sequences." Advances in neural information processing systems 33 (2020): 17283-17297.
>
> [11] Wang, Sinong, et al. "Linformer: Self-attention with linear complexity." arXiv preprint arXiv:2006.04768 (2020).
>
> [12] Choromanski, Krzysztof, et al. "Rethinking attention with performers." arXiv preprint arXiv:2009.14794 (2020).
>
> [13] Dao, Tri, et al. "Flashattention: Fast and memory-efficient exact attention with io-awareness." Advances in Neural Information Processing Systems 35 (2022): 16344-16359.
>
> [14] Dao, Tri. "Flashattention-2: Faster attention with better parallelism and work partitioning." arXiv preprint arXiv:2307.08691 (2023).
>
> [15] Li, Shenggui, et al. "Sequence parallelism: Long sequence training from system perspective." arXiv preprint arXiv:2105.13120 (2021).
>
> [16] Liu, Hao, Matei Zaharia, and Pieter Abbeel. "Ring Attention with Blockwise Transformers for Near-Infinite Context." arXiv preprint arXiv:2310.01889 (2023).
>
> [17] Li, Dacheng, et al. "LightSeq: Sequence Level Parallelism for Distributed Training of Long Context Transformers." arXiv preprint arXiv:2310.03294 (2023).
>
> [18] Korthikanti, Vijay Anand, et al. "Reducing activation recomputation in large transformer models." Proceedings of Machine Learning and Systems 5 (2023).
>
> [19] Jacobs, Sam Ade, et al. "DeepSpeed Ulysses: System Optimizations for Enabling Training of Extreme Long Sequence Transformer Models." arXiv preprint arXiv:2309.14509 (2023).
>
> [20] Bavarian, Mohammad, et al. "Efficient training of language models to fill in the middle." arXiv preprint arXiv:2207.14255 (2022).
>
> [21] Liu, Nelson F., et al. "Lost in the middle: How language models use long contexts." arXiv preprint arXiv:2307.03172 (2023).
>
> [22] Xiao, Guangxuan, et al. "Efficient streaming language models with attention sinks." arXiv preprint arXiv:2309.17453 (2023).

---

> > ### Comment · Reviewer_Dcc1 · 2023-11-16
> > **Re: Response**
> >
> > Thanks for the new experiments. My concerns have been solved and I raised my score to 8 as I promised.
> >
> > I sincerely suggest that please add the new experiments and related work to your draft no matter the paper is accepted or not. It is always good to make the work better, which should be irrelevant to the acceptance.
> >
> > One question: why the performance on long-range benchmarks is n-shaped? I mean, we can see the performance is going down when the base is very large. Any reason behind this?

---

### Official Review · Reviewer_jipE · 2023-10-30

**Soundness:** 3 good
**Presentation:** 3 good
**Contribution:** 2 fair
**Rating:** 6
**Confidence:** 5

**Summary:**

This paper focuses on the extrapolation capability of Rotary Position Embedding (RoPE) in large language models (LLMs). The authors first observe that pre-training RoPE with smaller or larger bases can significantly enhance its extrapolation performance. Then, they propose the Scaling Laws of RoPE-based Extrapolation, a unified framework that describes the relationship between extrapolation performance and base value as well as tuning context length from a periodic perspective. Furthermore, they explain the origin of the RoPE-based extrapolation issue by the critical dimension for extrapolation. Finally, they achieve extrapolation up to 1 million context length within only 16K training length on LLaMA2 7B and 13B.

**Strengths:**

- Observing that pre-training RoPE with smaller or larger bases can significantly enhance its extrapolation performance, providing a new perspective for extrapolation research.
- Proposing the Scaling Laws of RoPE-based Extrapolation, offering a unified theoretical framework for RoPE extrapolation.
- Explaining the origin of the RoPE-based extrapolation issue through the critical dimension for extrapolation.
- Achieving extrapolation up to 1 million context length within only 16K training length on LLaMA2 7B and 13B, demonstrating the potential of RoPE extrapolation.
- Providing guidance for other extrapolation methods, such as dynamic NTK and Code LLaMA.

**Weaknesses:**

- In fact, the analysis results of this article are basically available in the YaRN[4] paper, which had pointed out that certain dimensions have high frequency and enough turns are made during training, therefore, the training of these dimensions is sufficient and should remain unchanged during extrapolation; for those dimensions that are less than one circle, only interpolation (PI) can be used to ensure numerical stability. This is obviously the equivalent expression of the proposed concept "critical dimension" in this work.
- Most experimental comparisons only report ppl, which does not represent real long context ability. The authors should conduct experiments on some practical benchmarks and tasks. Eg. PASSKEY RETRIEVAL[2], the Hugging Face Open LLM Leaderboard[3] (ARC, HellaSwag, TruthfulQA, etc).
- The RoPE extrapolation methods discussed in the paper may have certain limitations in practical applications, it is just verified over perplexity, but not enough practical application scene.
- Compared to baselines, it lacks some comparison results over computational resources and memory usage.
- Missing reference. The work should compare results with ReRoPE[1].

[1] J. Su. Rectified rotary position embeddings. https://github.com/bojone/rerope, 2023.

[2] A. Mohtashami and M. Jaggi. Landmark attention: Random-access infinite context length for transformers, 2023. arXiv: 2305.16300.

[3] Hugging Face. Open LLM Leaderboard, 2023. URL https://huggingface.co/spaces/ HuggingFaceH4/open_llm_leaderboard.

[4] YaRN: Efficient Context Window Extension of Large Language Models

**Questions:**

- Only 7B and 13B models are verified. Have you experimented with the proposed methods on larger or smaller models? So that the "scaling law" saying can be more convincing.

---

> ### Author Response · Authors · 2023-11-15
> **Response (1/4)**
>
> Dear reviewer, thank you for your question. When it comes to the comparison with YaRN [1], we think our work is our concurrent work, because YaRN is published only one month before our submission. Compared with YaRN [1], our work gives a quantitative analysis of the critical dimension. Based on this analysis, we propose and validate the relationship between RoPE extrapolation, period, and attention score, and give the upper bound of model extrapolation under different fine-tuning bases. These can not be replaced and treated equally with simply saying that low-frequency dimensions can extrapolate directly and high-frequency dimensions need to be interpolated.
>
> When it comes to perplexity, perplexity is measured in all extrapolation work, such as Linear PI [2], CodeLLaMA [3] and LLaMA2Long [4]. Our work focuses on the curve of perplexity w.r.t. context length, which is commonly reported in related works [1-4].
>
> When it comes to the downstream tasks, our work actually provides the results of 0-shot NQ [5], TriviaQA [6], SuperGLUE [7] and 5-shot MMLU [8] in Appendix B.2. In order to present a comprehensive evaluation, we included new results on 0-shot Lambada [9], ARC-easy/challenge [10], Hellaswag [11], TruthfulQA [12], OpenbookQA [13], PIQA [14] and Winogrande [15]. We also show the average score over the above benchmark and updated 0-shot SuperGLUE [7] as well as 5-shot MMLU [8]. Particularly, based on the evaluation results, we find that the effect of tuning on the original context length as emphasized in our work is almost consistently better than tuning on the longer context length with the same base value. As a result, we achieve a balance between long context and short tasks by tuning with a larger base on the original length, which is a common issue raised in the work of expanding the context during the fine-tuning phase [1-4].

---

> ### Author Response · Authors · 2023-11-15
> **Response (2/4)**
>
> |  |  | lambada | SuperGLUE | MMLU | ARC-e | ARC-c | hellaswag | truthfulqa_mc | openbookqa | piqa | winogrande | average |
> | --- | --- | --- | --- | --- | --- | --- | --- | --- | --- | --- | --- | --- |
> | llama2_7B | pretrained | 73.30  | 50.43  | 46.78  | 50.26  | 36.61  | 69.45  | 35.09  | 32.00  | 76.66  | 61.25  | 53.18  |
> |  | ntk_fixed_8 | 67.53  | 51.13  | 39.36  | 48.68  | 34.58  | 68.43  | 35.09  | 29.40  | 76.28  | 61.48  | 51.20  |
> |  | finetuned | 72.62  | 51.49  | 47.73  | 54.32  | 38.31  | 69.60  | 35.53  | 38.00  | 75.95  | 61.40  | 54.49  |
> |  | finetuned_16k | 72.17  | 50.60  | 42.71  | 49.91  | 36.61  | 67.68  | 34.50  | 28.80  | 75.79  | 60.22  | 51.90  |
> |  | base_500 | 69.51  | 48.34  | 38.70  | 53.09  | 38.64  | 66.46  | 35.09  | 31.00  | 74.97  | 59.67  | 51.55  |
> |  | base_500_16k | 68.62  | 48.20  | 28.79  | 46.21  | 36.95  | 62.35  | 34.36  | 25.00  | 73.67  | 58.33  | 48.25  |
> |  | base_652 | 68.78  | 46.86  | 33.43  | 46.74  | 33.90  | 64.00  | 34.36  | 28.80  | 75.35  | 60.22  | 49.24  |
> |  | base_1304 | 69.24  | 47.22  | 33.46  | 49.38  | 37.63  | 65.51  | 35.09  | 25.00  | 76.06  | 60.30  | 49.89  |
> |  | base_2608 | 69.38  | 48.04  | 37.07  | 48.15  | 34.58  | 65.61  | 34.06  | 28.00  | 75.24  | 60.93  | 50.11  |
> |  | base_40000 | 72.77  | 51.02  | 47.34  | 55.03  | 36.95  | 69.46  | 35.23  | 36.20  | 76.06  | 61.33  | 54.14  |
> |  | base_40000_16k | 70.77  | 48.03  | 45.44  | 49.38  | 31.86  | 67.69  | 34.94  | 28.20  | 76.33  | 61.33  | 51.40  |
> |  | base_80000 | 72.68  | 51.20  | 46.87  | 53.09  | 37.63  | 69.44  | 35.23  | 37.40  | 76.01  | 60.69  | 54.02  |
> |  | base_80000_16k | 72.48  | 51.29  | 45.08  | 52.20  | 35.25  | 69.54  | 34.36  | 32.60  | 76.33  | 61.09  | 53.02  |
> |  | base_120000_16k | 72.66  | 50.34  | 43.75  | 51.85  | 34.24  | 69.54  | 34.21  | 32.20  | 76.22  | 61.96  | 52.70  |
> |  | base_160000 | 72.54  | 50.24  | 46.24  | 53.26  | 38.64  | 69.07  | 35.09  | 36.80  | 76.01  | 60.77  | 53.87  |
> |  | base_400000 | 72.25  | 51.08  | 46.01  | 52.03  | 38.64  | 68.86  | 35.23  | 36.00  | 75.68  | 60.85  | 53.66  |
> |  | base_600000 | 72.09  | 51.02  | 46.39  | 53.26  | 37.97  | 68.90  | 35.67  | 34.80  | 75.63  | 61.33  | 53.71  |
> |  | base_1000000 | 71.80  | 50.84  | 46.29  | 52.20  | 37.29  | 68.82  | 35.82  | 35.40  | 75.63  | 60.77  | 53.49  |
> |  | base_1000000_16k | 71.14  | 49.80  | 41.83  | 50.79  | 34.92  | 68.45  | 34.80  | 29.80  | 75.90  | 61.88  | 51.93  |
> | llama2_13B | pretrained | 76.48  | 52.35  | 55.77  | 59.44  | 40.00  | 74.22  | 34.65  | 42.60  | 79.54  | 63.85  | 57.89  |
> |  | ntk_fixed_8 | 72.56  | 52.64  | 52.04  | 58.38  | 40.34  | 73.56  | 34.80  | 37.00  | 79.71  | 62.67  | 56.37  |
> |  | finetuned | 75.12  | 53.19  | 55.78  | 62.79  | 42.03  | 75.19  | 35.09  | 55.00  | 79.00  | 62.98  | 59.62  |
> |  | finetuned_16k | 74.00  | 53.44  | 51.11  | 58.20  | 41.02  | 73.16  | 34.65  | 35.00  | 78.89  | 63.93  | 56.34  |
> |  | base_500 | 72.56  | 52.81  | 47.98  | 57.67  | 38.98  | 71.38  | 33.77  | 35.20  | 76.88  | 64.17  | 55.14  |
> |  | base_500_16k | 69.34  | 49.28  | 38.34  | 49.21  | 33.22  | 67.23  | 35.38  | 29.60  | 76.50  | 58.48  | 50.66  |
> |  | base_652 | 73.78  | 52.54  | 50.49  | 58.73  | 38.31  | 71.62  | 35.23  | 35.20  | 77.48  | 62.83  | 55.62  |
> |  | base_1304 | 74.21  | 52.19  | 53.59  | 58.73  | 41.36  | 72.38  | 35.23  | 36.80  | 77.37  | 64.01  | 56.59  |
> |  | base_2608 | 74.54  | 52.54  | 55.06  | 60.67  | 42.71  | 73.63  | 34.80  | 43.40  | 78.02  | 63.22  | 57.86  |
> |  | base_40000 | 74.87  | 53.11  | 55.26  | 60.49  | 42.37  | 74.76  | 35.53  | 52.60  | 79.60  | 62.98  | 59.16  |
> |  | base_40000_16k | 75.06  | 54.08  | 53.68  | 59.61  | 41.69  | 74.33  | 35.53  | 38.00  | 78.78  | 62.51  | 57.33  |
> |  | base_80000 | 74.89  | 52.37  | 55.32  | 61.20  | 41.36  | 74.76  | 35.38  | 53.80  | 79.27  | 62.90  | 59.12  |
> |  | base_80000_16k | 74.77  | 53.28  | 52.86  | 61.38  | 38.98  | 75.11  | 34.94  | 38.20  | 79.05  | 63.61  | 57.22  |
> |  | base_120000_16k | 75.06  | 53.91  | 52.93  | 61.90  | 41.02  | 74.85  | 34.65  | 43.60  | 78.94  | 63.61  | 58.05  |
> |  | base_160000 | 74.89  | 52.63  | 54.35  | 60.14  | 42.03  | 74.50  | 35.96  | 54.20  | 79.05  | 62.75  | 59.05  |
> |  | base_400000 | 74.23  | 52.47  | 54.01  | 62.96  | 42.37  | 74.50  | 35.82  | 54.40  | 78.89  | 62.19  | 59.18  |
> |  | base_600000 | 74.19  | 54.14  | 53.75  | 61.20  | 42.03  | 74.24  | 35.67  | 53.60  | 79.00  | 63.77  | 59.16  |
> |  | base_1000000 | 74.58  | 50.82  | 52.94  | 60.85  | 42.37  | 74.36  | 35.53  | 51.80  | 78.67  | 62.98  | 58.49  |
> |  | base_1000000_16k | 74.71  | 51.95  | 51.61  | 59.44  | 39.66  | 74.68  | 35.53  | 40.80  | 79.11  | 63.30  | 57.08  |

---

> ### Author Response · Authors · 2023-11-15
> **Response (3/4)**
>
> We also compared some tasks with 16K context length used in long context research [1,16]. On one hand, concating multi-shot short prompts into a long context [1,4] is included, such as 25-shot ARC-challenge [1,10], 10-shot Hellaswag [1,11], and 16-shot GSM8K [17] in LEval [16]. On the other hand, QA and extraction tasks of long documents are also involved, such as TOEFL-QA [18] and TopicRetrieval [19] in L-Eval [16]. Our method is still capable of these tasks, as shown in the table below. Particularly, even for RoPE-based LLMs fine-tuned on pre-training context, namely 4k tokens, they still have certain effects on these tasks with 16K context size. This also proves one of the claims of our work, that tuning with the original pre-training context can also boost the length extrapolation of RoPE-based LLMs. We will publish more evaluation results for long document tasks in the later version.
>
> |  |  | 25-shot ARC-c | 10-shot hellaswag | 10-shot GSM | TOEFL-QA | topic_retrieval |
> | --- | --- | --- | --- | --- | --- | --- |
> | llama2_7B | pretrained | 48.14  | 72.78  | 0.00  | 34.94  | 0.00  |
> |  | ntk_fixed_8 | 46.78  | 68.94  | 5.00  | 34.20  | 31.33  |
> |  | ntk_dynamic | 48.14  | 72.73  | 9.00  | 37.17  | 24.67  |
> |  | base_10000 | 47.80  | 72.49  | 0.00  | 18.59  | 0.00  |
> |  | base_80000 | 47.12  | 72.66  | 11.00  | 36.43  | 29.33  |
> |  | base_160000 | 45.42  | 72.60  | 10.00  | 40.15  | 38.00  |
> |  | base_400000 | 47.12  | 72.63  | 8.00  | 41.64  | 36.67  |
> |  | base_600000 | 46.10  | 72.54  | 12.00  | 38.66  | 34.00  |
> |  | base_1000000 | 46.78  | 72.71  | 9.00  | 31.97  | 34.00  |
> | llama2_13B | pretrained | 54.58  | 73.89  | 0.00  | 59.48  | 0.00  |
> |  | ntk_fixed_8 | 50.85  | 66.88  | 8.00  | 58.74  | 36.00  |
> |  | ntk_dynamic | 54.58  | 73.93  | 22.00  | 59.85  | 30.00  |
> |  | base_10000 | 53.56  | 75.53  | 0.00  | 56.13  | 0.00  |
> |  | base_80000 | 53.56  | 75.69  | 20.00  | 63.94  | 30.00  |
> |  | base_160000 | 53.56  | 75.50  | 23.00  | 62.83  | 39.33  |
> |  | base_400000 | 54.24  | 75.56  | 17.00  | 59.48  | 34.67  |
> |  | base_600000 | 53.90  | 75.29  | 14.00  | 63.94  | 36.00  |
> |  | base_1000000 | 52.54  | 75.18  | 10.00  | 64.68  | 28.67  |

---

> ### Author Response · Authors · 2023-11-15
> **Response (4/4)**
>
> When it comes to the comparison of computational resources and memory usage, since we only modify the rotation angle base of RoPE, no additional computation and memory are added. At the same time, since we conduct the training with an original context length, 4K, to achieve 100K context length, compared with Linear PI [2], CodeLLaMA [3], YaRN [1], and LLaMA2Long [4], our work also has certain advantages in the fine-tuning speed and memory cost. If the paper gets accepted, we will add detailed data about these contents. Thanks to the reviewer for the reminder.
>
> Finally, when it comes to the reference of ReRoPE [20] and the verification of larger (such as 70B) and smaller (such as 330M, 1B) RoPE-based LM, we are conducting the verification of the 70B model at present. Since the computational cost is huge, we will add the results to our paper in the later version.
>
> [1] Peng, Bowen, et al. "Yarn: Efficient context window extension of large language models." arXiv preprint arXiv:2309.00071 (2023).
>
> [2] Chen, Shouyuan, et al. "Extending context window of large language models via positional interpolation." arXiv preprint arXiv:2306.15595 (2023).
>
> [3] Roziere, Baptiste, et al. "Code llama: Open foundation models for code." arXiv preprint arXiv:2308.12950 (2023).
>
> [4] Xiong, Wenhan, et al. "Effective long-context scaling of foundation models." arXiv preprint arXiv:2309.16039 (2023).
>
> [5] Kwiatkowski, Tom, et al. "Natural questions: a benchmark for question answering research." Transactions of the Association for Computational Linguistics 7 (2019): 453-466.
>
> [6] Joshi, Mandar, et al. "Triviaqa: A large scale distantly supervised challenge dataset for reading comprehension." arXiv preprint arXiv:1705.03551 (2017).
>
> [7] Wang, Alex, et al. "Superglue: A stickier benchmark for general-purpose language understanding systems." Advances in neural information processing systems 32 (2019).
>
> [8] Hendrycks, Dan, et al. "Measuring massive multitask language understanding." arXiv preprint arXiv:2009.03300 (2020).
>
> [9] Paperno, Denis, et al. "The LAMBADA dataset: Word prediction requiring a broad discourse context." arXiv preprint arXiv:1606.06031 (2016).
>
> [10] Clark, Peter, et al. "Think you have solved question answering? try arc, the ai2 reasoning challenge." arXiv preprint arXiv:1803.05457 (2018).
>
> [11] Zellers, Rowan, et al. "Hellaswag: Can a machine really finish your sentence?." arXiv preprint arXiv:1905.07830 (2019).
>
> [12] Lin, Stephanie, Jacob Hilton, and Owain Evans. "Truthfulqa: Measuring how models mimic human falsehoods." arXiv preprint arXiv:2109.07958 (2021).
>
> [13] Mihaylov, Todor, et al. "Can a suit of armor conduct electricity? a new dataset for open book question answering." arXiv preprint arXiv:1809.02789 (2018).
>
> [14] Bisk, Yonatan, et al. "Piqa: Reasoning about physical commonsense in natural language." Proceedings of the AAAI conference on artificial intelligence. Vol. 34. No. 05. 2020.
>
> [15] Sakaguchi, Keisuke, et al. "Winogrande: An adversarial winograd schema challenge at scale." Communications of the ACM 64.9 (2021): 99-106.
>
> [16] An, Chenxin, et al. "L-Eval: Instituting Standardized Evaluation for Long Context Language Models." arXiv preprint arXiv:2307.11088 (2023).
>
> [17] Cobbe, Karl, et al. "Training verifiers to solve math word problems." arXiv preprint arXiv:2110.14168 (2021).
>
> [18] Tseng, Bo-Hsiang, et al. "Towards machine comprehension of spoken content: Initial TOEFL listening comprehension test by machine." arXiv preprint arXiv:1608.06378 (2016).
>
> [19] Li, Dacheng, et al. How long can open-source llms truly promise on context length?, 2023. URL https://lmsys.org/blog/2023-06-29-longchat.
>
> [20] Su, Jianlin. Rectified rotary position embeddings., 2023. https://github.com/bojone/rerope.

---

> ### Public Comment · ~Yingsheng_Wu1 · 2023-11-17
> **Experimental results of a 110M(gpt-2 base, but use RoPE) model for reference**
>
> Settings：
>
> model_size = 110M
>
> n_layers = 12
>
> embed_dim = 768
>
> train_max_length = 1024
>
> text_max_length = 4096
>
> epochs = 5
>
> datasets = wikitext-103
>
> others as same as gpt2-base
>
> ------------------------------------------------
> Result (best in 5 epochs):
>
> base=100 PPL=79.72
>
> base=1000 PPL=32.87
>
> base=10000 PPL=159.79
>
> base=100000 PPL=270.94
>
> base=1000000 PPL=320.32

---

> > ### Author Response · Authors · 2023-11-17
> >
> > Thank you for your interest in our work and your comments. Regarding the phenomenon you report, I hope you can further report on some of your training configurations. Is it fine-tuning with a different base value as discussed in this article, or directly changing the base value in the inference stage or the pre-training stage? I am still not clear about the changing process of perplexity w.r.t. the length of language modeling. In addition, regarding the laws we propose, there may indeed be a range of applicable model sizes, and even those scaling laws that are generally accepted still have such a phenomenon. Thank you for reporting these results on the 110M size model. It has some reference significance for us. We will do the same in the future. We are also doing some experiments on the 70B size model.

---

> ### Comment · Reviewer_jipE · 2023-11-21
>
> Thank you very much for the sincere reply!
>
> The author's response can address some of my questions, and I hope the author can incorporate these experiments into the next version of the manuscript.
>
> I agree that this work has discovered some rope-based patterns for long context extrapolation, which will be helpful to research in this direction.
>
> However, I believe there's potential for enhancement. For instance, the term "scaling law" is employed, yet the largest evaluation model merely reaches 13 billion. Furthermore, numerous studies have focused on optimizing Llama in long context scenarios, as cited in the authors' response. However, there's a scarcity of evidence demonstrating their tangible impact on practical applications.
>
> Therefore, I decided to improve my score to 6.

---

### Official Review · Reviewer_Gdv1 · 2023-11-02

**Soundness:** 4 excellent
**Presentation:** 4 excellent
**Contribution:** 4 excellent
**Rating:** 10
**Confidence:** 4

**Summary:**

Rotary Position Embeddings have played a significant role in improving the scalability of LLMs. However, RoPE faces problems when trying to extrapolate beyond the length the model was trained on.

Various approaches have been taken to solve the issue with extrapolation, and this paper goes in depth describing why these issues exist and derives pretty good solutions as well. Apart from this, it presents a good understanding of why some of the existing methods work and why a base of 10,000 is not a good value for RoPE's base.

All the claims are very well substantiated with theoretical as well as empirical analysis.

**Strengths:**

- Pretty useful theorems that shows why and how RoPE's beta impacts extrapolation.
- Results on critical base, d_{extra}, and T_{extra} and their relationship is a very useful contribution in this area.
- Both experimental and theoretical analysis are strong.

**Weaknesses:**

A bit more clarification around the calculation of critical beta, and why extrapolation capabilities exist even when beta values are large could be added since we will see OOD embeddings as soon as T is larger than T_{train} during inference.

**Questions:**

None.

---

> ### Author Response · Authors · 2023-11-15
>
> Dear reviewer, thank you very much for your recognition and comments on this paper. Concerning the clarification of the critical base and the extrapolation of a larger base beyond T_{train}, we will add the following discussion if the paper gets accepted. I don’t know whether the following discussion can make you feel OK.
>
> The critical base refers to a base value that exists when extending the further training context. This base value is just enough to make the period of the original critical dimension equal to the length of the further training context length. In this way, RoPE-based LLM only uses the dimensions within the original critical dimension to perceive the position information that happens to be within the continued training length. If the base is smaller, then more dimensions can sense position information, and the perplexity curve will be flatter; if the base is larger, then longer position information can be represented by dimensions within the critical dimension, thereby expanding the supported context window.
>
> When it comes to the extrapolation beyond T_{train}, this is still related to the critical dimension. For dimensions within the critical dimension, the inputs of sin and cos representing the positional embedding can traverse 0 to 2$\pi$, and any value can be expressed normally; for the dimensions beyond the critical dimension, the inputs of sin and cos have not even completed a round. After enlarging the base, the rotation angle $\theta$ will shrink. For the same sin and cos values, the allowed input position index will become larger.
> For the dimensions beyond the critical dimension, when the value of the position index * $\theta$ exceeds the value seen during training, then ood embedding will occur, resulting in extrapolation problems; while the value of the input subscript * $\theta$ still falls within the range seen at the pre-training stage, there will be no extrapolation problem. In other words, the increase in base leads the rotation angle to become smaller, which increases the range of allowed values for the index in the position encoding, thus enabling extrapolation.

---

### Meta-Review · Area_Chair_5gyp · 2023-12-05

**Metareview:**

This paper presents an in-depth study on the extrapolation capabilities of LLMs using Rotary Position Embedding (RoPE), focusing on how to adjust the rotary base and enhance extrapolation performance. This approach is underpinned by both theoretical and empirical analyses, including experiments with 1 million context length in testing based on a 16K training length.

The reviewers have generally given positive feedback and pointed out strengths of the paper:
- The paper is well written and studies an interesting problem.
- The theoretical results provide insightful analysis on the rotary base in RoPE and length extrapolation.
- The experimental results are good.

The reviewers also raised concerns regarding evaluation metrics, and the authors provided solid additional experiments to address the concerns. All the reviewers give positive ratings to this paper. Therefore, we recommend acceptance.

**Justification For Why Not Higher Score:**

While the paper provides robust theoretical and experimental evidence for its claims, the scope of its empirical evaluation is somewhat limited. Experiments are only conducted on one model family (LLaMA) and two sizes (7B and 13B). Besides, a reviewer’s follow-up question on the n-shaped performance curve is not addressed.

**Justification For Why Not Lower Score:**

The paper offers both theoretical insights and practical solutions on length generalizable Transformers. In particular, the findings of how RoPE can be optimized for better extrapolation performance are useful to the community. This, along with the positive responses from the reviewers, warrants its acceptance.

---

### Decision · Program_Chairs · 2024-01-16

Accept (poster)